# Puzzle: Distillation-Based NAS for Inference-Optimized LLMs

**Akhiad Bercovich** [*]  **Tomer Ronen** [*]  **Talor Abramovich  Nir Ailon  Nave Assaf  Mohammad Dabbah  Ido Galil
Amnon Geifman  Yonatan Geifman  Izhak Golan  Netanel Haber  Ehud Karpas  Roi Koren  Itay Levy
Pavlo Molchanov  Shahar Mor  Zach Moshe  Najeeb Nabwani  Omri Puny  Ran Rubin  Itamar Schen
Ido Shahaf  Oren Tropp  Omer Ullman Argov  Ran Zilberstein  Ran El-Yaniv**

## Abstract

Large language models (LLMs) offer remarkable capabilities, yet their high inference costs restrict wider adoption. While increasing parameter counts improves accuracy, it also broadens the gap between state-of-the-art capabilities and practical deployability. We present *Puzzle*, a hardware-aware framework that accelerates the inference of LLMs while preserving their capabilities. Using neural architecture search (NAS) at a large-scale, Puzzle optimizes models with tens of billions of parameters. Our approach utilizes blockwise local knowledge distillation (BLD) for parallel architecture exploration and employs mixed-integer programming for precise constraint optimization.

We showcase our framework's impact via Llama-3.1-Nemotron-51B-Instruct (Nemotron-51B) and Llama-3.3-Nemotron-49B, two publicly available models derived from Llama-70B-Instruct. Both models achieve a $2.17\times$ inference throughput speedup, fitting on a single NVIDIA H100 GPU while retaining 98.4% of the original model's benchmark accuracies. These are the most accurate models supporting single H100 GPU inference with large batch sizes, despite training on 45B tokens at most, far fewer than the 15T used to train Llama-70B. Lastly, we show that lightweight alignment on these derived models allows them to surpass the parent model in specific capabilities. Our work establishes that powerful LLM models can be optimized for efficient deployment with only negligible loss in quality, underscoring that inference performance, not parameter count alone, should guide model selection.

---

[*]Equal contribution . Correspondence to: Akhiad Bercovich, Tomer Ronen, Ran El-Yaniv <{abercovich, tronen, relyaniv}@nvidia.com>.

*Proceedings of the 42nd International Conference on Machine Learning*, Vancouver, Canada. PMLR 267, 2025. Copyright 2025 by the author(s).

## 1. Introduction

With remarkable advancements in the capability and accuracy of LLMs, these models are increasingly adopted in various domains, ranging from virtual assistants to sophisticated enterprise solutions. This adoption trend is accompanied by a growing appetite for larger and more powerful models, as evidenced by the industry's push toward increasingly larger-scale LLMs (OpenAI, 2023; Anil et al., 2023; cla; Dubey et al., 2024) and Inference-time compute scaling (GPT-o1, (Wei et al., 2022; Yao et al., 2023; Hardt & Sun, 2024)). However, the high computational costs associated with these models and their projected future iterations – particularly during inference – restrict their accessibility and scalability, thus presenting a significant challenge for widespread personal and commercial applications.

LLMs require a substantial amount of parameters for their training process to converge easily and achieve better generalization (Kaplan et al., 2020; Hoffmann et al., 2024; Allen-Zhu et al., 2019; Belkin et al., 2018). This over-parameterization not only facilitates optimization, but also provides greater capacity to store knowledge and learn complex patterns across diverse tasks, explaining why larger models consistently demonstrate superior performance (Kaplan et al., 2020; Hoffmann et al., 2024). However, once trained, many parameters and computations turn out to be redundant for inference, as evidenced by the success of computational efficiency techniques (He et al., 2024; Beltagy et al., 2020; Hu et al., 2022). Yet, LLM architectures remain largely uniform, comprising repeated identical layers, with little consideration given to balancing each block's computational cost against its contribution to overall model predictive performance—a design choice primarily driven by training stability and ease of scaling rather than inference efficiency. This work addresses how to transform a trained LLM from a structure suited for training into one optimized for efficient inference on specific hardware (such as H100), while preserving its accumulated knowledge and predictive performance. Given a "*parent model*", our approach explores a large search space of architecture configurations to identify efficient options tailored to meet specific hardware and task-related constraints. This exploration requires

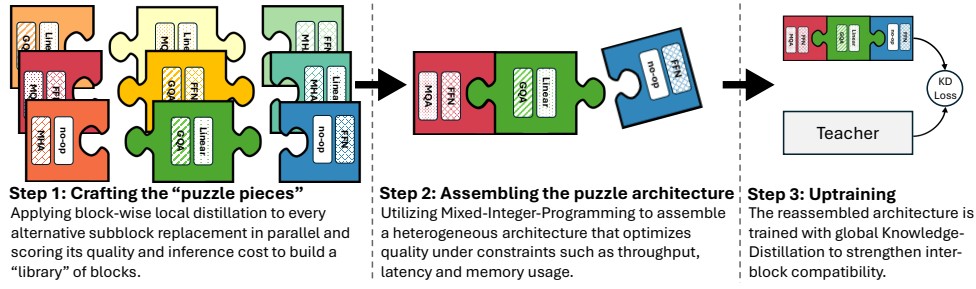

**Step 1: Crafting the "puzzle pieces"**
Applying block-wise local distillation to every alternative subblock replacement in parallel and scoring its quality and inference cost to build a "library" of blocks.

**Step 2: Assembling the puzzle architecture**
Utilizing Mixed-Integer-Programming to assemble a heterogeneous architecture that optimizes quality under constraints such as throughput, latency and memory usage.

**Step 3: Uptraining**
The reassembled architecture is trained with global Knowledge-Distillation to strengthen inter-block compatibility.

*Figure 1.* An overview of the three stages of our Puzzle framework.

a method to reliably estimate the performance of each potential configuration, allowing us to identify models that balance efficiency and accuracy for deployment.

In this paper, we introduce the *Puzzle* framework, summarized in Figure 1, which pioneers a decomposed *neural architecture search* (NAS) strategy for LLMs to explore and optimize a vast search space of possible model architectures for the target hardware. Inspired by methods in computer vision NAS, and LANA (Molchanov et al., 2022) especially, we define design spaces that include alternative attention and feed-forward network (FFN) layers of varying efficiency degrees, up to a complete layer skip in the extreme case. We then use a *blockwise local distillation* (BLD) (See Section 3) framework to train all these block variants for all layers of the parent LLM in parallel. Next, we efficiently score each alternative replacement "puzzle piece" and search an enormous design space for the most accurate models, while adhering to a set of inference constraints (e.g., memory size, latency, and throughput). This is done by using a *Mixed-Integer-Programming* (MIP) algorithm. Lastly, the reassembled model is trained with *Global Knowledge Distillation* (GKD) (Hinton et al., 2015). Unlike traditional uniform transformer architectures, our NAS framework produces non-uniform models with adapted computation allocation, optimizing each layer's expressivity based on the model's overall requirements to focus resources where they matter most. This leads to significant gains in efficiency without compromising model expressivity. By focusing on parent models with SOTA performance, we derive child models pushing the efficient frontier (see Figure 5 and Table 5), e.g., models which provide the best accuracy per dollar.

Our framework offers several advantages. First, it enjoys extremely low costs relative to training a model from scratch. For instance, the entire training—BLD before the MIP stage and GKD afterward—required 45B tokens to run on Llama-3.1-70B-Instruct (henceforth Llama-70B), compared to more than 15T tokens used to train the parent model. Even smaller budgets are possible—down to 4B tokens—while still preserving strong performance (see Table 6). Additionally, our method requires only the parent model's weights—not its training data—making it ideal for

"open-weights, closed-data" scenarios where training data of the parent model is not publicly available. This allows practitioners to take any freely-available model and tailor it to their specific hardware or use case. To demonstrate the effectiveness of our framework, we present Llama-3.1-Nemotron-51B-Instruct (Nemotron-51B), derived from the Llama-70B parent model using Puzzle. Nemotron-51B breaks the efficient frontier of LLMs on a single NVIDIA H100 GPU, establishing a new state-of-the-art for commercial applications by achieving unmatched throughput and memory efficiency on this hardware. Interestingly, the Nemotron-51B resulting architecture is unique and irregular, with many layers featuring reduced or skipped attention and FFN operations. This design enhances NVIDIA H100 utilization under FP8 quantization while preserving accuracy. We also introduce a derivative of Llama-3.1-8B-Instruct, which breaks the efficient frontier for its throughput slice. While Nemotron-51B also leads within its parameter range, we argue that categorizing models solely by parameter count—such as 50B or 70B—is inadequate for real-world applications. Instead, inference performance under specific hardware, inference engine, quantization levels, budget constraints, and usage profiles—such as varying sequence lengths and batch sizes—should guide model selection.

Conventional approaches to designing LLMs for inference, such as training from scratch or knowledge distillation, present significant challenges. Training a model from scratch is impractical for evaluating multiple configurations. Knowledge distillation, while generally faster due to guidance from a teacher model, remains prohibitively costly when evaluating multiple candidates from a large search space. Puzzle circumvents these limitations by conducting architecture search immediately after BLD, during the search stage (MIP, Stage 2 in Figure 1). These stages efficiently identify promising configurations for multiple levels (slices) of inference constraints, without requiring GKD for each candidate. The computationally intensive GKD process is reserved for the final stage, after the optimal architectures have been reassembled. This enables Puzzle to focus resources on refining a single optimized model for each slice while maintaining low overall costs.

**Our contributions**: (1) We introduce Puzzle, a framework that applies decomposed NAS to distill an LLM into hardware and inference scenario optimized models. Our framework flexibly supports optimization across ranges of multiple constraints, including throughput, latency, and memory usage. Our work pioneers the large-scale use of blockwise distillation and MIP-based architecture search for LLMs, successfully scaling these techniques to tens of billions of parameters while requiring only a fraction of the original training compute. We are the first to present NAS-derived models at the scale of tens of billions of parameters. Puzzle is designed to be low-cost, enabling the efficient creation of multiple child models from a single parent LLM, each tailored to different hardware and inference requirements. This scalability makes it feasible to publish optimized variants for diverse use cases and deployment environments. A demo of Puzzle is available at: https://github.com/NVlabs/puzzle/

(2) Using Puzzle, we introduce **Nemotron-51B** and **Nemotron-49B**, both optimized for a single H100 GPU, thus setting a new benchmark for commercial applications. We show Nemotron-49B preserves its parent's 128K context via short uptraining and benefits from RLHF alignment—outperforming Nemotron-51B on some tasks. Puzzle's robustness is demonstrated throughout the paper by applying it numerous times with varied constraints, datasets, budgets and parent models.

(3) Our method focuses on optimization for real world scenarios, running on actual inference engines and efficient quantization levels (FP8). We therefore enhance TensorRT-LLM to efficiently support non-uniform blocks and attention mechanisms with varying numbers of key-value heads across layers. This results in models that can directly improve the costs and usability of running LLM inference in practical applications (see Appendix C).

(4) We provide a comprehensive empirical analysis of the relationship between architectural choices and hardware efficiency, offering insights to guide the design of future hardware-aware LLM architectures.

## 2. Search Space

The motivation for applying NAS in our work lies in the redundancy found in many trained LLMs. Numerous studies have shown that a significant portion of computations and parameters in LLMs become redundant post-training, during inference (Sanyal et al., 2024; Hu et al., 2022; Aghajanyan et al., 2021). Prior studies tried to solve these issues with techniques such as pruning (Muralidharan et al., 2024; Men et al., 2024; Ma et al., 2023; Xia et al., 2024; 2022), removing entire layers (He et al., 2024; Gromov et al., 2024), local attention methods (e.g., window and sink attention) (Beltagy et al., 2020; Xiao et al., 2024), reducing the number of

key-value (KV) heads (Shazeer, 2019; Ainslie et al., 2023) and many more. Given the high computational and monetary cost associated with running LLMs, optimizing these models to eliminate inefficiencies becomes a critical goal. NAS methods are defined by the search space, the search strategy and evaluation metric of candidate architectures. NAS offers a systematic approach to exploring architectural changes that balances performance and resource constraints.

We define a vast search space that encompasses different operations for each layer of the parent LLM model. A *block* is composed of smaller components called *subblocks*. While blocks are user-defined, in LLMs, a block typically refers to a transformer layer, with the subblocks being the attention module and the feed-forward network (FFN). For each transformer layer $i$, the search space combines options for the attention subblock (denoted $\mathcal{A}_i = \{a_j\}_{j=1}^m$, where $m$ is the number of possible attention subblocks) and FFN subblock (denoted $\mathcal{F}_i = \{f_k\}_{k=1}^n$ for $n$ subblocks). The attention subblock options could include mechanisms such as standard multi-head attention (MHA), *grouped query attention* (GQA) (Ainslie et al., 2023) with varying numbers of key-value heads, replacing the attention layer with a single linear layer, or no-op (i.e., entirely skipping the subblock). The FFN options include full or reduced intermediate dimensions (which could be obtained with pruning techniques), linear layers, or no-ops. The combined search space for each transformer layer (or parent block), represented as $\mathcal{A}_i \times \mathcal{F}_i$, captures all possible pairings of attention and FFN configurations. In this work, each parent transformer layer can be replaced by a single corresponding child transformer layer (block), although theoretically, multiple parent layers could be grouped and replaced by a different number of child blocks (i.e., $P$ parent layers to $L$ child blocks). Specifically: (1) **Attention subblocks**: For $\mathcal{A}_i$, we consider different variants of GQA with varying numbers of key-value heads (8, 4, 2, and 1). We also include the option to replace the entire attention subblock with a single linear layer or skip it entirely using a no-op operation. (2) **FFN subblocks**: For $\mathcal{F}_i$, we consider the full intermediate-dimension expansion factor of the parent model, along with reduced dimensions of approximately: 87%, 75%, 50%, 25%, 20% and 10% of the original intermediate size. Furthermore, linear layers and no-op options are also included.

These variants offer a tradeoff between memory efficiency, computational cost, and representational power, allowing for flexibility based on specific resource constraints and performance needs. For example, reducing the number of key-value heads (by using GQA with fewer heads) not only speeds up the attention computation, but also helps decrease memory usage by reducing the KV-cache size, which can be crucial for meeting memory constraints or enabling larger batch sizes for better throughput (as GPUs operate more efficiently with larger batches). Using linear layers or re-

duced FFN dimensions can similarly lower computational requirements, whereas keeping full dimensions maintains higher representational power for better accuracy. Lastly, in this work, we require that all subblocks within a layer have the same input and output dimensions as their parent. However, a scheme with subblocks of varying embedding dimensions could be developed in future work.

To illustrate the scale of the search space, consider Llama 3.1-70B (Dubey et al., 2024), a model with 80 transformer layers. For the specific instantiation of our framework presented in this work, we defined each transformer layer to have 6 potential alternatives for the attention subblock and 9 alternatives for the FFN subblock, resulting in 54 possible configurations per layer. Consequently, the total number of potential child model architectures is $54^{80}$, which is approximately $10^{138}$ different architectures. This number greatly exceeds the estimated number of atoms in the observable universe ($10^{82}$). Given such an immense number of possibilities, and considering the costs of partially training or even measuring the accuracy of a single LLM architecture, evaluating any representative subset of the search space is computationally infeasible. Therefore, designing a traditional search strategy and evaluation scheme for NAS in such a vast space is challenging. To address this, we devised an efficient decomposed local distillation and evaluation framework, and our decomposed search strategy (described in Section 3 and Section 4). These strategies allow feasible navigation of the search space to find configurations that balance expressivity and efficiency with practical constraints like latency, memory usage, and throughput.

## 3. Blockwise Local Distillation

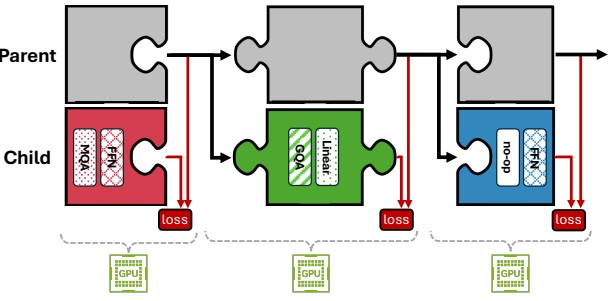

*Figure 2.* BLD: blocks are trained in parallel and independently.

To create capable "puzzle pieces"-a set of trained block variants forming a *block library* for architectural exploration-we need to set effective weights for each child block. Our method involves efficient and training-free initializations for these blocks (see Appendix A). While these initialization techniques are beneficial for low-budget experiments, performance can be significantly improved by distilling knowledge from each parent block to its corresponding child block.

Our approach decomposes the crafting process to operate on individual blocks rather than full child models, which drastically decreases the computational cost. Since gradients do not need to propagate across blocks, each child block can be trained independently and in parallel to locally mimic its corresponding parent block. Only the parent activations are transferred between layers, as illustrated in Figure 2. This local distillation approach offers several advantages. First, because each child block relies solely on its corresponding parent block, activations and gradients are isolated from other child blocks. This independence enables training blocks separately, leveraging pipeline parallelism across multiple GPUs. Second, each child subblock is trained to mimic a relatively simple function—a single parent subblock—making the process considerably simpler and more stable than training an entire child model. This focused training facilitates faster convergence and allows higher learning rates compared to standard language modeling or GKD methods. Additionally, we find that this approach requires only a small dataset (approximately one billion tokens). Third, each child subblock benefits from high-quality outputs from its preceding parent subblock, rather than the lower-quality outputs typical in global model training, which further enhances convergence speed.

To optimize the performance of each child block, we feed parent activations into the current block and compute a normalized mean squared error (MSE) loss (Kurtic et al., 2023). Specifically, we define the loss as $\mathcal{L} = \frac{\text{MSE}(o_p, o_c)}{\text{MSE}(o_p, 0)}$, where $o_p$ and $o_c$ represent the outputs of the original parent block and the modified child block, respectively.

A primary limitation of BLD is that it does not ensure compatibility between different blocks. This issue arises because each block is trained with inputs from the preceding parent blocks rather than from the outputs of its predecessor blocks within the child model. This prevents later blocks from adapting to the errors of earlier blocks, and may lead to errors propagating and compounding through the child model. To mitigate this, we introduce GKD as a final training phase in our framework (see Section 4.3). Nonetheless, empirical results show that **the BLD stage alone recovers much of the parent model's performance** (see Table 16).

To ensure broad coverage of diverse data domains within limited training schedules, we curated a dataset mixture, termed *Distillation Mix*, for all our distillation training runs. This mixture includes source code repositories, Wikipedia articles, books, news websites, and several other domains. The dataset comprises 224 billion tokens collected from three public datasets: FineWeb (Penedo et al., 2024), Dolma (Soldaini et al., 2024), and Buzz-V1.2 (Hive-Digital-Technologies). In our BLD experiments, we used 1 billion training tokens. We discuss the effect of varying BLD training lengths on downstream tasks in Appendix F.1.3.

### 3.1. Building a Block Library with Decoupled Blockwise Local Distillation

The first stage of our decomposed NAS framework (further discussed in Section 4) is building a "library" of trained blocks. In order to cover the entire search space defined in Section 2, we need to obtain trained weights for each attention variant $a_j$ and each FFN variant $f_k$ in each transformer layer $i$. We consider two methods to train the block library: *coupled BLD* and *decoupled BLD*. For each transformer layer $i$, coupled BLD constructs each possible block variant $[a_j, f_k]_i$ and trains it to emulate the corresponding parent block $[a_{\text{parent}}, f_{\text{parent}}]_i$. Given the significantly higher computational costs of LLMs compared to CV models, and noting the inherent structure of the transformer layer, we propose decoupled BLD to drastically reduce the cost of building a block library: training $[a_j, f_{\text{parent}}^{(\text{frozen})}]_i$ and $[a_{\text{parent}}^{(\text{frozen})}, f_k]_i$ separately to emulate $[a_{\text{parent}}, f_{\text{parent}}]_i$ while freezing the child subblock that is identical to the parent, and composing the trained subblocks into a full block $[a_j, f_k]_i$ after training.

Given $m$ attention variants, $n$ FFN variants, and $l$ transformer layers, coupled BLD requires training $m \cdot n \cdot l$ variants, while decoupled BLD requires only $(m+n) \cdot l$ variants, significantly speeding up library construction, visualized in Figure 3. This efficiency becomes especially critical with a large toolbox for $\mathcal{A}_i$ and $\mathcal{F}_i$. For instance, with $m = n = 20$ and $l = 80$ layers, decoupled BLD would require training 3,200 blocks compared to 32,000 for coupled BLD. Decoupled BLD enables us to explore a large search space and produce high quality models. In Appendix F.1.1 we present a technique to combine coupled BLD and decoupled BLD.

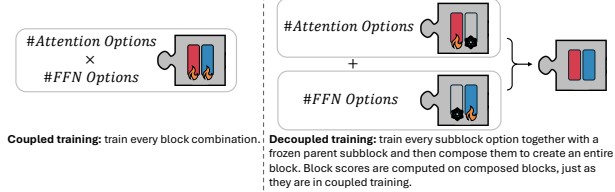

**Coupled training:** train every block combination. **Decoupled training:** train every subblock option together with a frozen parent subblock and then compose them to create an entire block. Block scores are computed on composed blocks, just as they are in coupled training.

*Figure 3.* Coupled BLD requires training $|\mathcal{A}_i| \times |\mathcal{F}_i|$ variants per transformer layer, while decoupled BLD requires only $|\mathcal{A}_i| + |\mathcal{F}_i|$ variants per layer, significantly speeding up library construction.

## 4. Decomposed Search Algorithm for LLMs

Our decomposed NAS framework is similar to earlier decomposed NAS methods used in computer vision (CV) such as DNA (Li et al., 2019), DONNA (Moons et al., 2020), and LANA (Molchanov et al., 2022): (1) **Build a block library:** construct a diverse block library using BLD (see Section 3.1). (2) **Estimate block resource requirements:** estimate the runtime and memory requirements of each block variant across different scenarios. (3) **Score blocks:** score each block variant across the network to estimate its quality relative to the parent block in that location. (4) **Search architectures:** use a search algorithm to construct "Puzzle architectures" that have the best estimated quality under specified runtime and memory constraints. While these steps mirror established NAS approaches, their application to LLMs presents unique challenges due to the massive scale of the models. Our innovations in scaling these techniques to multi-billion-parameter models are detailed in the following subsections.

### 4.1. Estimating Resource Requirements

Accurate estimation of computational costs is crucial for optimizing LLM architectures for real-world deployment. While theoretical metrics like FLOPs or parameter count are commonly used to approximate cost, they often fail to capture the complexities of hardware acceleration. The actual runtime of neural network operations depends on numerous hardware-specific factors: the number of streaming multiprocessors (SMs), tensor cores, memory bandwidth, and I/O patterns all significantly impact performance. For instance, operations with fewer FLOPs might run slower in practice due to poor hardware utilization. This disparity between theoretical and actual performance makes direct measurement on target hardware essential for optimization.

Memory requirements during inference comprise two distinct components with different scaling behaviors. Parameter memory, while substantial, remains constant regardless of the input size. In contrast, the key-value cache memory scales linearly with both batch size and sequence length, often becoming the dominant factor in long-sequence scenarios. For example, in a model with 32 attention heads and 128-dimensional head size, each token requires 8KB of KV-cache per layer using FP16 precision. For an 8K-token sequence with batch size 64, this amounts to 4GB per layer for the KV-cache, possibly exceeding parameter memory.

LLM inference has two phases with different performance characteristics. The prefill phase processes the input with high parallelization, using fewer passes than decoding. While efficient, the computational cost is significant for long contexts. In contrast, the generation phase processes one query token at a time auto-regressively, requiring repeated forward passes, and often employing paged attention to manage memory efficiently. This difference between phases makes it crucial to actually measure both prefill and generation runtime in the target scenarios (see Table 2).

Batch size plays a critical role in hardware efficiency, especially during the generation phase. During prefill, even small batches process substantial amounts of data due to sequence length, allowing efficient hardware utilization. However, generation with small batch sizes processes minimal data per layer while still performing all the IO needed to load the layer parameters, causing severe hardware under-utilization.

Increasing the batch size creates larger tensors that better utilize the GPU, often significantly improving throughput.

Given these complexities, our approach measures resource requirements directly on target hardware across various scenarios. For each block variant, we collect prefill and generation latencies across different sequence lengths and batch sizes at a chosen quantization level. These measurements define the constraints in our MIP optimization (Appendix B), enabling the search algorithm to find architectures optimized for actual deployment rather than in theory.

### 4.2. Scoring Architecture Solutions

A key advantage of decomposed NAS is its ability to estimate the quality of an assembled model based on metrics gathered from its individual blocks. This capability allows search algorithms to explore an enormous search space very efficiently, as the quality of each candidate architecture encountered during the search can be estimated within less than a second, instead of having to actually realize the candidate model and calculate some measure that requires performing forward passes on the entire model such as validation accuracy. Traditional NAS methods in CV, such as those in (Zoph & Le, 2016) and (Real et al., 2017), typically rely on full model evaluation, which is computationally prohibitive for LLMs, where both search spaces and inference costs are substantially larger.

We score each block variant at each network location by measuring the impact of replacing only that specific block in the parent model. To do this, we construct a model identical to the parent but with a single block replaced by a trained block from our library, then calculate a performance measure on the entire model. For efficient I/O, when scoring multiple variants, we load onto the GPU only the blocks that differ from the previously evaluated model. We call these scores *replace-1-block scores*. During architecture search, we estimate the quality of a constructed architecture as the sum of its individual replace-1-block scores (see Appendix B). Note that scoring is not performed on candidate models during the search phase – we estimate the quality of each block only once, then use its replace-1-block score to estimate the quality of any candidate that contains it.

We consider several metrics as potential replace-1-block scores: *(1) downstream accuracy:* accuracy on downstream tasks such as MMLU (Hendrycks et al., 2021). While downstream accuracy is popular in CV applications, it can be very costly to measure in LLMs, especially given the number of block variants requiring scoring. *(2) LM loss:* causal language modeling loss, defined as the average log-likelihood of a validation corpus under the model's next-token prediction distribution. *(3) KL divergence:* Kullback–Leibler divergence between the next-token prediction distributions of the evaluated model and the parent model, averaged across

tokens in a validation corpus. KL divergence is widely used in KD setups as a statistical distance measure but has not been explored for decomposed NAS scoring before. Our analysis in Appendix F.1.4 highlights its effectiveness.

**Search Algorithm: Mixed-integer Programming.** We treat the process of choosing a block variant for each transformer layer as a grouped Knapsack problem, imposing constraints on memory (parameters plus key-value cache), throughput (tokens generated per second), and latency (time per batch). Specifically, we employ a mixed-integer programming (MIP) formulation that maximizes a global quality score while satisfying these hardware-driven requirements. This approach allows us to efficiently navigate enormous search spaces (on the order of $\sim 10^{138}$ possible architectures) and find solutions tailored to the usage needs and target hardware. Optimization details, including the formal problem definition, are provided in Appendix B.

### 4.3. Post-Puzzle Inter-Block Uptraining

As mentioned in Section 3, our BLD step trains each child block on parent-produced distributions, not on the outputs of the preceding child block. Consequently, block compatibility may be suboptimal post-BLD. To address this, we perform a short end-to-end *global knowledge distillation (GKD)* phase, also called Knowledge Distillation, in which the child (student) aligns with the parent (teacher) (Hinton et al., 2015; Muralidharan et al., 2024; Lu et al., 2022). In particular, we minimize:

$$\mathcal{L}_{\text{GKD}} = \underbrace{\sum_{l=1}^{L}\big(1 - \frac{\mathbf{h}_c^l \cdot \mathbf{h}_p^l}{\|\mathbf{h}_c^l\|\|\mathbf{h}_p^l\|}\big)}_{(1)\ \text{Cosine Similarity Loss}} + \underbrace{\sum_{i=1}^{N} p_i \log\big(\frac{p_i}{q_i}\big)}_{(2)\ \text{KL Divergence Loss}}, \quad (1)$$

where (1) measures how well the student matches the teacher's representations, and (2) aligns output distributions.

We find that explicitly including language-modeling loss here is not beneficial, as it can degrade downstream tasks (see Appendix F.3.1). The effectiveness of this GKD step is further demonstrated in Appendix F.3.

## 5. Main Results

Using our Puzzle framework, we generated Nemotron-51B as a child derivative of the Llama-70B model. Nemotron-51B achieves a significant improvement in inference efficiency while retaining nearly all the accuracy of its parent, demonstrating the effectiveness of our approach.

**Evaluating Model Performance:** To evaluate Puzzle-derived child models like Nemotron-51B, two performance metrics are of primary interest: 1) **Accuracy Preservation**: This measures how much of the parent model's accuracy is retained by the child. A high retention percentage indicates

that Puzzle produces high-performing children. 2) **Computational Efficiency**: This reflects the child model's ability to adhere to the constraints it was optimized for. In our case, the focus is on *throughput*, showing how we improved the model's suitability for reducing inference cost.

These metrics demonstrate the balance between model quality and achieving optimization for deployment needs.

**Accuracy comparison:** Table 1 compares the accuracy of Nemotron-51B with its parent across several benchmarks. On average, Nemotron-51B retains 98.4% of its parent's accuracy, even exceeding it on certain benchmarks such as MT-Bench and GSM8K Chat. This underscores the robustness of the Puzzle framework in maintaining model quality despite **significant** architectural modifications.

*Table 1.* Accuracy comparison of Nemotron-51B with Llama-70B across several benchmarks. Accuracy preserved is the ratio of child to parent accuracy. *Chat prompt as defined in Adler et al. (2024). **version by CodeParrot.

| Benchmark | Llama-3.1-70B-Instruct | Nemotron-51B | Accuracy Preserved (%) |
|---|---|---|---|
| Winogrande (Sakaguchi et al., 2020) | 85.08 | 84.53 | 99.35 |
| ARC Challenge (Clark et al., 2018) | 70.39 | 69.20 | 98.30 |
| MMLU (Hendrycks et al., 2021) | 81.66 | 80.20 | 98.21 |
| HellaSwag (Zellers et al., 2019) | 86.44 | 85.58 | 99.01 |
| GSM8K (Cobbe et al., 2021) | 92.04 | 91.43 | 99.34 |
| TruthfulQA (Lin et al., 2022) | 59.86 | 58.63 | 97.94 |
| XLSum English (Hasan et al., 2021) | 33.86 | 31.61 | 93.36 |
| MMLU Chat* | 81.76 | 80.58 | 98.55 |
| GSM8K Chat* | 81.58 | 81.88 | 100.37 |
| Instruct HumanEval (n=20) (Chen et al., 2021)** | 75.85 | 73.84 | 97.35 |
| MT-Bench (Zheng et al., 2023) | 8.93 | 8.99 | 100.67 |

**Human evaluation:** We complement the above accuracy performance benchmarks with a human evaluation comparing the parent and child models. A blind test comparison of the models was conducted on a general purpose test set that includes tasks such as reasoning, longform text generation, knowledge and more. Results show comparable performance between the models (Figure 4), strengthening the claim that the accuracy degradation is minimal. For more details about the evaluation see Appendix D.

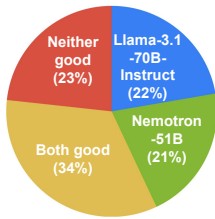

*Figure 4.* In a blind test, human annotators rated Llama-70B and Nemotron-51B comparably.

**Throughput comparison:** Table 2 specifies the throughput performance of Nemotron-51B against its parent across diverse input-output sequence lengths. Nemotron-51B achieves speedups up to 2.17x, enabling larger workloads per GPU and making it highly efficient for deployment. For each model and hardware configuration, we automatically

selected the optimal TP and batch size to maximize throughput per GPU. The inference engine handled this selection dynamically for each run. For example, Nemotron-51B achieved optimal throughput with TP=1 and batch size 256, while Llama-3.1-70B performed best with TP=4 and batch size 384.

*Table 2.* Throughput comparison of Nemotron-51B and Llama-70B across various scenarios. Throughput is measured in tokens per second per GPU (NVIDIA H100). TP# indicates the number of GPUs used in tensor parallelism. Note: Results were obtained on NVIDIA H100 SXM GPUs with FP8 quantization for weights, activations and KV cache using TensorRT-LLM. Optimal tensor parallelism was used for each model. Input/output sequence lengths indicate the prefill (input) and decode (output) operations performed by the LLM. *TP=1 is not the optimal configuration for Llama-3.1-70B and is included for equal-resource comparison.

| Scenario | Input/Output | Nemotron-51B (TP#) | Llama-3.1-70B-Instruct (TP#) | Speedup |
|---|---|---|---|---|
| Chatbot | 128/128 | 5478 (TP1) | 2645 (TP1) | 2.07 |
| Text Generation | 128/1024 | 6472 (TP1) | 2975 (TP4) / 1274 (TP1*) | **2.17 / 5.08** |
| Long Text Generation | 128/2048 | 4910 (TP2) | 2786 (TP4) | 1.76 |
| Inference-time compute | 128/4096 | 3855 (TP2) | 1828 (TP4) | 2.11 |
| Summarization/RAG | 2048/128 | 653 (TP1) | 339 (TP4) / 301 (TP1*) | 1.92 / 2.17 |
| Stress Test | 2048/2048 | 2622 (TP2) | 1336 (TP4) | 1.96 |

**Accuracy vs. throughput frontier:** The tradeoff between accuracy and efficiency is key for model selection, impacting deployment costs. Nemotron-51B is designed to balance the two and push beyond the current efficient frontier. Because throughput directly affects cost, Nemotron-51B provides the best accuracy per dollar, as shown in Figure 5. To account for both knowledge and conversational capabilities, accuracy is measured as a weighted combination of MMLU and MT-Bench scores: (MT-Bench $\times$ 10 + MMLU) / 2.

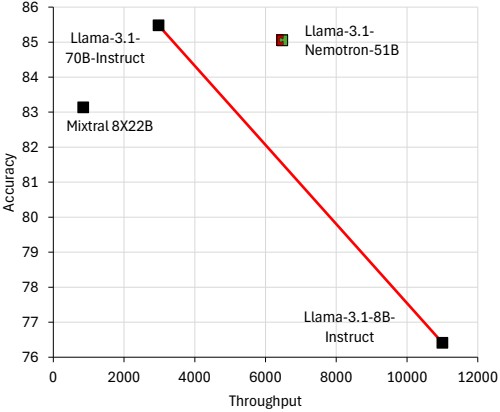

*Figure 5.* Accuracy vs. Throughput performance of Nemotron-51B compared to state-of-the-art models. Throughput measured on NVIDIA H100 GPUs with optimal TP setting, running in FP8 on a "text generation" scenario (see Table 2). The red line represents the efficient frontier, highlighting models with the best accuracy-to-throughput tradeoff. Accuracy=(MT-Bench $\times$10 + MMLU) / 2

The Nemotron-51B architecture achieves substantial computational savings through strategic reduction of computation

across many layers, as shown in Figure 6. Observing the figure it is evident that our framework discovered significant optimization opportunities in both early layers (0-15) and later layers (45-70), with computed savings shown in green. Different regions exhibit distinct optimization patterns: early layers show balanced reduction in attention and FFN components, while later layers demonstrate more aggressive attention optimization. Notably, the framework identifies a central region (layers 16-42) that maintains full computational capacity, suggesting these middle layers are critical for performance. This automatically discovered structure demonstrates that large efficiency gains are achievable through careful targeting of computational reduction, while maintaining computation where it matters most.

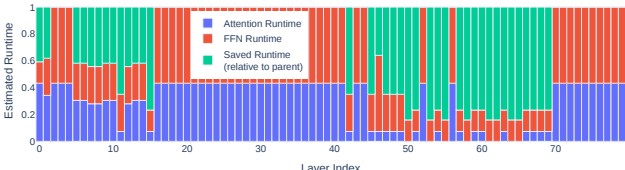

*Figure 6.* The runtime of the attention and FFN subblocks of Nemotron-51B, relative to the original Llama-70B subblocks.

### 5.1. Additional Evaluations and Puzzle derivatives

**Long-Context Performance:** We evaluated Nemotron-51B on a subset of the RULER benchmark (Hsieh et al., 2024). Notably, although Nemotron-51B was trained only on sequences up to 8K tokens, it retained over 96% of its parent's accuracy at 16K tokens, showing Puzzle's ability to preserve performance beyond the direct training length. Performance degraded beyond 64K tokens, suggesting that fine tuning on longer contexts could extend the effective context length.

To demonstrate this extension, we produced *Llama-3.3-Nemotron-49B-Super-Base* (henceforth Nemotron-49B-Base, the base version of Llama-3.3-Nemotron-Super-49B[1]) from Llama-3.3-70B-Instruct with identical constraints to Nemotron-51B and uptrained on an additional 5B tokens at 64K context and 5B at 128K. Table 3 shows that Nemotron-49B-Base maintains or exceeds its parent's performance up to 16K tokens, retains over 98% at 64K, and remains above 94% at 128K. Full performance details are provided in Appendix E.

**Alignment:** We further aligned *Nemotron-49B-Base* by following the RLHF and instruction-tuning recipe of Wang et al. (2024), using 25M tokens. Table 4 compares results before and after alignment, showing that Puzzle derivatives can undergo RLHF-based alignment to boost their accuracy.

*Table 3.* Comparison of Llama-3.3-70B-Instruct (parent) and Nemotron-49B-Base (child) on RULER for context lengths up to 128K.

| Context | Parent Average Score | Child Average Score | Accuracy Preserved (%) |
|---|---|---|---|
| 4K | 96.77 | 97.40 | **100.65** |
| 8K | 96.46 | 96.59 | **100.13** |
| 16K | 95.98 | 96.09 | **100.11** |
| 32K | 94.70 | 94.30 | **99.57** |
| 64K | 88.91 | 87.39 | **98.29** |
| 128K | 52.25 | 49.62 | **94.96** |

*Table 4.* The performance of Nemotron-49B-Base before and after alignment, even surpassing its parent in certain benchmarks.

| Model | MMLU | MT-Bench | Arena Hard (Li et al., 2024) |
|---|---|---|---|
| Nemotron-49B-Base (after alignment) | 80.98 | 9.21 | 82.1 |
| Nemotron-49B-Base (before alignment) | 81.03 | 8.77 | 65.8 |
| Llama-3.3-70B-Instruct (parent) | 81.66 | 8.84 | 71.70 |

**Compact model:** We applied Puzzle to produce a child derivative of Llama-3.1-8B-Instruct, optimized for throughput specifically on an RTX 4090 GPU. This model breaks the efficient frontier for its throughput range, demonstrating Puzzle's ability to deliver highly efficient architectures also on consumer-grade hardware, while preserving the balance between accuracy and performance. Table 5 highlights this model's superior tradeoff in accuracy and efficiency.

*Table 5.* Accuracy and throughput of our high-throughput child derivative of Llama-3.1-8B-Instruct, which achieves equivalent throughput to Llama-3.2-3B-Instruct and far better accuracy. Throughput is estimated via the sum of measured block runtimes on a single NVIDIA RTX 4090 GPU, measured with an input-output sequence length of 1024 tokens each, the scenario for which this model was optimized. Accuracy = (MT-Bench $\times 10$ + MMLU) / 2.

| Model | Throughput* | Accuracy |
|---|---|---|
| Ours (child) | 5856 | 73.98 |
| Llama-3.2-3B-Instruct | 5737 | 70.36 |
| Llama-3.1-8B-Instruct (parent) | 3385 | 76.40 |

**Additional Puzzle derivatives:** Beyond the aforementioned models, Puzzle was also used in the development of the following derivatives: (1) Puzzle was applied to Llama-3.1-405B-Instruct with constraints requiring a $1.5\times$ latency speedup and compatibility with a single NVIDIA $8\times$H100 node (640 GB) or a single B100 GPU (192 GB). The resulting model retained 99.5% of the parent model's accuracy (averaged across MMLU, MT-Bench, MMLU-Pro, HumanEval, and Arena Hard). *FFN Fusion* (Bercovich et al., 2025) was subsequently applied to further improve latency, followed by continued pretraining—resulting in Llama-3.1-Nemotron-Ultra-253B-CPT, which serves as the base for the publicly released Llama-3.1-Nemotron-Ultra-253B [2]. (2) In Blakeman et al. (2025), a variation of Puzzle—referred to as

---

[1] https://huggingface.co/nvidia/Llama-3_3-Nemotron-Super-49B-v1

[2] https://huggingface.co/nvidia/Llama-3_1-Nemotron-Ultra-253B-v1

"miniPuzzle"—was applied to Nemotron-H-56B-Base under constraints targeting RTX 5090 deployment with a 1M context window. The resulting 47B model retained 99.94% of the parent model's accuracy (averaged over MMLU, MMLU-Pro, GSM8K, and HellaSwag). The key distinction between miniPuzzle and Puzzle is that miniPuzzle enforces a homogeneous block design across all layers.

**Training Token Budget:** The GKD phase described earlier utilized 45B tokens, which might exceed practical budgets for some users. However, in practice, our method can achieve substantial accuracy recovery with significantly fewer tokens. Our experiments, summarized in Table 6, demonstrate that notable accuracy recovery can be realized with reduced token usage. After only 3.7B tokens of GKD, Nemotron-51B recovered 98.8% of its parent's accuracy on MMLU and MT-Bench benchmarks. Similarly, Nemotron-49B regained 99.63% of its parent's accuracy after only 8.68B tokens, and even 98.47% after just 2.9B tokens. These results indicate that while 45B tokens are modest compared to full model training from scratch, users can flexibly adjust GKD token counts based on available resources without severely compromising accuracy.

*Table 6.* Effectiveness of GKD training with reduced token budgets. The table shows the performance recovery relative to the parent model for Nemotron-51B and Nemotron-49B using various amounts of tokens in GKD.

| Model | GKD Tokens (B) | Performance (MMLU / MT-Bench) | Accuracy Preserved (%) |
|---|---|---|---|
| Nemotron-49B-Base | 8.68 | 80.73 / 8.87 | 99.63 |
| Nemotron-49B-Base | 2.9 | 80.72 / 8.675 | 98.47 |
| Nemotron-49B-Base | 0.72 | 80.4 / 8.59 | 97.79 |

### 5.2. Ablation Studies Highlights:

We performed a series of ablations to examine how local distillation, block search strategies, data composition, and search space design each contribute to Puzzle's accuracy–efficiency tradeoffs. See Appendix F for full details. Below we summarize some of the most interesting findings:

(1) **Decoupled BLD:** A decoupled construction of the block library—training block alternatives independently—significantly reduces cost by converting the search space from multiplicative to additive. This enables the construction of a large block library at a fraction of the compute cost. Interestingly, adding a coupled training pass over a narrowed subspace of promising combinations provides an additional performance boost, suggesting hybrid BLD strategies can be beneficial (Appendix F.1.1).

(2) **Data Composition and Scale:** We studied the effects of dataset scale and composition on BLD quality. Accuracy improves with more tokens, but saturates beyond 0.5B tokens. Moreover, domain-diverse data improves robustness; however, training on even a narrow domain retains over 93% of full performance, highlighting the robustness of block-

level supervision and indicating diminishing returns past moderate data sizes (Appendices F.1.2, F.1.3).

(3) **Block Scoring Metrics:** We evaluated various scoring metrics to guide architecture search. KL divergence generally yields the best results across tasks, outperforming LM loss and task-specific metrics. However, in specialized settings, such as instruction tuning benchmarks, task-grounded scores may offer an advantage. These results emphasize the importance of choosing scoring functions aligned with target deployment scenarios (Appendix F.1.4).

(4) **Search Space Diversity:** We examined the impact of restricting the search space. Replacing all layer blocks with their original counterparts or limiting substitutions to no-op variants significantly degrades performance. This confirms that diversity in architectural choices is critical for discovering high-quality configurations and validates the necessity of block-level flexibility in Puzzle (Appendix F.1.5).

(5) **MIP Global Search vs. Greedy and Random Baselines:** We compared MIP-based global search with a simpler greedy algorithm that selects blocks layer-wise under fixed budgets. Despite meeting the same throughput constraints, greedy search yields significantly worse architectures, highlighting the need for joint optimization across layers. Additional baselines, such as random-from-library selection, show that naïve strategies even using trained blocks fall well short of MIP (Appendix F.2.2, F.2.4).

## 6. Conclusions and Future Directions

In this work we present Puzzle, a framework that modifies LLMs from their over-parameterized training configurations to optimized, inference-efficient architectures tailored for specific hardware. Puzzle achieves these improvements with remarkable efficiency in training resources. Requiring fewer than 50B tokens—compared to the trillions needed to train models from scratch—Puzzle produces high-performing models at a fraction of the usual cost. This extreme search and training efficiency still results in drastic inference performance improvements of Puzzle optimized models.

The success of Puzzle opens several promising directions for future research. Our introduction of decoupled BLD, which is significantly more efficient than coupled BLD, makes it feasible to evaluate a much larger set of potential blocks within a single optimization run. This efficiency enables exploration of novel operations as alternative Puzzle blocks, such as variable window attention mechanisms (Beltagy et al., 2020), *state-space models* (Gu et al., 2022; Gu & Dao, 2024), or other architectural innovations. The framework could also be extended to optimize models for specific capabilities, such as Chain-of-Thought reasoning or multimodal tasks, including vision-language models (Liu et al., 2023) and retrieval-augmented generation (Lewis et al., 2020).

## Impact Statement

The Puzzle framework's ability to adapt state-of-the-art models to hardware constraints, while maintaining high performance, can help make powerful AI systems more affordable and broadly deployable across diverse use cases. This broader adoption could expand the societal benefits of generative AI but also amplify its inherent risks (Solaiman et al., 2023; Capraro et al., 2024; Baldassarre et al., 2023). At the same time, by reducing energy consumption and hardware requirements needed for inference, Puzzle may alleviate some of the resource-intensive challenges of large-scale AI deployments, contributing to a more sustainable approach to generative AI.

## Acknowledgments

We thank Saurav Muralidharan and Sharath Turuvekere Sreenivas for fruitful discussions.

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

## A. Initialization of Alternative Subblocks

To accelerate the distillation process, we introduce training-free initialization techniques for alternative subblocks. We propose a method to reduce the intermediate dimensions of FFN subblocks by selectively pruning channels based on their contribution. Our approach, called *Channel Contribution*, uses an activation-based strategy to estimate channel contribution during forward passes over a calibration dataset, similar to (Muralidharan et al., 2024). To guide pruning, we rank intermediate channels based on their impact on the FFN output. This is quantified as the distance between the original FFN output and the output after pruning. The channels with the lowest contributions are prioritized for pruning during subblock initialization.

To formalize our method, let us denote the hidden dimension by $H$, the FFN intermediate dimension by $I$, the FFN down projection matrix by $W^{\text{down}} \in \mathbb{R}^{I \times H}$. Let $X \in \mathbb{R}^I$ represent the FFN's intermediate activations for a single token. The output of the FFN, $Y \in \mathbb{R}^H$, is then given by

$$Y = (W^{\text{down}})^\top X = \sum_{k=1}^{I} X_k W^{\text{down}}_{k,:}$$

We define the per-token contribution of channel $i$ as:

$$C_i(X) = \left\| \left( \sum_{k=1}^{I} X_k W^{\text{down}}_{k,:} \right) - \left( \sum_{\substack{k=1 \\ k \neq i}}^{I} X_k W^{\text{down}}_{k,:} \right) \right\|_2$$

$$= |X_i| \cdot \left\| W^{\text{down}}_{i,:} \right\|_2$$

We then compute the average contribution of each channel across all tokens in the calibration dataset.

When replacing an FFN subblock with a linear layer, we initialize it by computing the product of the up and down projection matrices for FFNs, thereby effectively ignoring the gating mechanism.

In attention subblocks with a reduced number of key-value heads, we initialize the projection matrices for the key and value heads by mean-pooling them into single projection matrices, following the approach used in (Ainslie et al., 2023). When replacing an attention subblock with a linear layer, we use the product of the value and output projection matrices, which simulates the scenario where each token attends only to itself.

## B. Search Algorithm: Mixed-integer Programming

The search space for LLM architecture optimization is enormous (we consider $\sim 10^{138}$ possibilities), as discussed in

Section 2. Efficiently navigating this space requires two key components: a fast method to estimate candidate quality, which we addressed in Section 4.2, and an efficient algorithm to maximize the estimated quality under hardware-specific constraints.

The structure of transformer models naturally frames our optimization problem as a grouped variant of the classical *Knapsack Problem*. Each layer of the model represents a group, containing various block alternatives (attention and FFN variants) as items. Each block alternative has an associated value (its quality score) and multiple costs (parameter memory, KV-cache memory, and runtime characteristics). The objective is to select exactly one block variant from each layer while maximizing the total score and satisfying deployment constraints.

Following (Molchanov et al., 2022), we formulate this as a Mixed Integer Programming (MIP) problem. Let $x_{i,j}$ be a binary decision variable indicating whether block variant $j$ is selected for layer $i$. For a model with $L$ layers and $K_i$ variants per layer, the optimization problem is:

$$\text{maximize} \quad \sum_{i=1}^{L} \sum_{j=1}^{K_i} \text{score}(i, j) x_{i,j}$$

$$\text{subject to} \quad \sum_{i=1}^{L} \sum_{j=1}^{K_i} [\text{mem}_{\text{params}}(i, j) + b \cdot \text{mem}_{\text{kv}}(i, j)] x_{i,j} \leq \text{Memory}_{\text{max}}$$

$$\frac{b \cdot \text{seq\_len}}{\sum_{i=1}^{L} \sum_{j=1}^{K_i} \text{runtime}(i, j, b) x_{i,j}} \geq \text{Throughput}_{\text{min}}$$

$$\sum_{i=1}^{L} \sum_{j=1}^{K_i} \text{runtime}(i, j, b) x_{i,j} \leq \text{Latency}_{\text{max}}$$

$$\sum_{j=1}^{K_i} x_{i,j} = 1 \quad \forall i \in \{1, \ldots, L\}$$

$$x_{i,j} \in \{0, 1\} \quad \forall i, j \,,$$

where:

- $\text{score}(i, j)$ is the quality score of block variant $j$ in layer $i$, measuring how well it maintains the parent model's performance. If the block scores represent a negative impact (e.g. LM loss or KL divergence) we minimize the sum of scores instead of maximizing it.

- $\text{mem}_{\text{params}}(i, j)$ is the parameter memory required for block variant $j$ in layer $i$, which is shared across all sequences in a batch.

- $\text{mem}_{\text{kv}}(i, j)$ is the key-value cache memory required for a single sequence in layer $i$ with block variant $j$.

- $b$ is the batch size - the number of sequences processed in parallel during inference.

- $\text{seq\_len}$ is the total sequence length, including both prefill and generation.

- $\text{runtime}(i, j, b)$ is the runtime of block variant $j$ in layer $i$ when processing batch size $b$.

- $\text{Memory}_{\text{max}}$ is the maximum allowed total memory, specified to fit the target GPU(s).

- $\text{Throughput}_{\text{min}}$ is the minimum required throughput (tokens per second).

- $\text{Latency}_{\text{max}}$ is the maximum allowed latency per batch.

The objective maximizes the sum of quality scores across all selected block variants. The first constraint ensures the total memory usage stays within limits, accounting for both parameter memory (shared across batches) and KV-cache memory (which scales linearly with batch size as each sequence requires its own cache). The second constraint enforces a minimum throughput requirement: for batch size $b$, we process $b \cdot \text{seq\_len}$ tokens within the total runtime, which must meet or exceed $\text{Throughput}_{\text{min}}$ tokens per second. The third constraint ensures the total processing time for a batch does not exceed the maximum allowed latency. The fourth constraint guarantees exactly one variant is selected for each layer. Note that we do not impose any constraint on the number of model parameters - they are reduced naturally by the search algorithm due to the true scenario constraints.

Since the batch size $b$ is not a variable in this optimization, we solve the MIP problem multiple times with different values of $b$ to explore the runtime-memory trade-off space. Larger batch sizes typically result in higher throughput for all block operations, but also higher latency and more memory for KV-cache storage, which forces a reduction in memory to meet the constraints (such as reducing the number of KV heads). For each set of deployment constraints, we choose the batch size that produced the highest quality architectures. If the target scenario specifies a maximum batch size, such as the typical number of active users for a chat bot, the search can be capped accordingly.

While MIP problems are NP-complete, modern solvers can efficiently handle instances of our size. Using the open-source `python-mip` package (Inc., 2023), we obtain high-quality solutions within seconds. This efficiency enables us to explore multiple architecturally diverse solutions by adding a diversity constraint:

$$\sum_{i=1}^{L} \sum_{j=1}^{K_i} x_{i,j} y_{i,j} \leq \alpha \cdot L \quad \forall y \in Y,$$

where $Y$ is the set of previous solutions and $\alpha \in [0, 1]$ controls the maximum allowed similarity. For example, with $\alpha = 0.8$, each new solution must differ from previous solutions in at least 20% of its layer choices. This constraint helps discover meaningfully different architectures.

A key feature of our approach is its ability to generate solutions precisely tailored for specific hardware platforms. For example, in platforms with limited memory intended for batch size 1, the algorithm strongly favors memory-saving techniques such as FFN pruning, while de-prioritizing KV-cache optimizations which have minimal impact at batch 1. The hardware specificity extends to architectural features of the inference devices - for example, on H100 GPUs, FP8 quantization offers $\sim 2\times$ acceleration and can be used aggressively, while on A100 GPUs where FP8 is unavailable, different optimization strategies must be employed. Even the difference in inter-GPU bandwidth between H100 PCI-E and NVLink configurations influences the optimal architecture by affecting tensor parallel synchronization costs.

This flexibility enables a powerful "train once, adapt on demand" methodology that requires minimal human intervention. After building a block library and computing block scores once, we can efficiently generate different architectures optimized for various deployment scenarios without additional training or manual tuning. The user needs only to specify the available block configurations for each platform - the hardware-specific measurements of block variants naturally guide the optimization toward platform-appropriate solutions. This approach makes Puzzle particularly valuable for real-world deployment, where the same parent model might need to be optimized differently across diverse hardware configurations and deployment constraints.

## C. Supporting Fast Inference for Variable-Block Architectures in TensorRT-LLM

TensorRT-LLM is a highly optimized LLM runtime designed to accelerate inference performance on NVIDIA GPUs. It provides industry-leading performance for both latency and throughput-oriented workloads. The runtime enables LLMs to run on GPUs while utilizing custom paged attention (Kwon et al., 2023) kernels to efficiently manage KV caching across sequences in a batch. Furthermore, TensorRT-LLM supports FP8 quantization and various scheduling policies that allow LLM deployments to optimally utilize the underlying hardware.

To enable the use of Puzzle-generated architectures from the search space (see Section 2) a major underlying assumption of TensorRT-LLM had to be revised: that all attention layers contain the same number of key and value heads. We devised changes to the paged KV cache strategy that enabled variable GQA ratios within a model. TensorRT-LLM now supports running any architecture from our search space including variable blocks and linear replacements, using FP8 precision for weights, activations and KV cache. Our NAS framework is designed to generate models that run efficiently in real inference scenarios. Therefore full support in

inference engines and awareness to runtime considerations when applying on NAS scheme contributes greatly to the usability of resulting models, such as Nemotron-51B.

## D. Human Evaluation

A blind-test comparison between Nemotron-51B and Llama-70B was conducted in the following manner:

- A set of 169 samples was sent.

- The evaluation was done by Nvidia's data factory team.

- Three annotators annotated each sample independently, resulting in a total of $169 \cdot 3 = 507$ annotations.

- Annotators saw [prompt, completion 1, completion 2] and had to choose between 4 options:
    - Completion 1 is better.
    - Completion 2 is better.
    - Both are good.
    - Neither is good.

- The order of the completions was randomized to avoid positional bias.

The test set was curated by the project's product team and Nvidia's data factory and included the following tasks and subtasks listed in brackets: The test set was curated and included the following tasks and subtasks listed in brackets:

- Long form text generation (write a blog, write a story, other).

- Long inputs (Open book QA, Multi-hop questions, text to headline)

- Grounded text generation (table of contents to blog, paraphrasing).

- Multi-condition instructions (3-5 conditions, 6-10 conditions).

- Knowledge and trivia.

- Summarization (full document summary, summarize to bullet points, summarize paragraph to a sentence).

- Reasoning (temporal reasoning, cause and effect, navigation, general reasoning).

- Semantic extraction.

## E. RULER Benchmark Performance Tables

This section provides complete performance tables for our parent-child pairs on a subset of the RULER benchmark across all context lengths evaluated.

### E.1. Nemotron-51B vs. Llama-3.1-70B-Instruct

Table 19 shows the results of the parent (Llama-3.1-70B-Instruct) and child (Nemotron-51B) for all context lengths. As noted in the main text, Nemotron-51B was trained on sequences up to only 8K tokens yet retains more than 96% of its parent's performance at 16K. The child's performance degrades at 64K and beyond, which is unsurprising given its training horizon. Nevertheless, these results underscore that a large fraction of the parent's long-context capabilities can remain intact even without explicit training on such long sequences.

### E.2. Nemotron-49B-Base vs. Llama-3.3-70B-Instruct

In Table 7, we present the extended context-length results for Nemotron-49B-Base (uptrained with sequences up to 128K tokens) alongside its parent (Llama-3.3-70B-Instruct). We rename columns to make them more consistent with the style above, although some tasks differ from those used for Nemotron-51B. Nemotron-49B-Base preserves 98% or more of its parent's performance up to 64K tokens and remains above 94% at 128K. This highlights that adding a short uptraining phase on longer contexts can effectively extend the context range of Puzzle-derived models. Additional details and per-task results appear below.

## F. In-Depth Analysis and Ablation Studies

To better understand the key components and design choices of the Puzzle framework, we conduct a series of detailed analyses and ablation studies. We evaluate the importance of global knowledge distillation, investigate the impact of training dataset size and composition, and analyze how our MIP solver adaptively chooses architectures under varying constraints. These studies not only validate our design decisions but also provide insights into the fundamental trade-offs in LLM architecture optimization and the relative importance of different architectural components.

### F.1. Block Library Construction Ablation Studies

In Appendices F.1.1 to F.1.5, we analyze the impact of critical decisions in block library construction. These studies examine the fundamental trade-offs between computational cost, dataset characteristics, and model performance. Our key findings are:

- **Coupled vs. Decoupled BLD:** Decoupled BLD reduces training cost by transforming the search space from multiplicative to additive. Combining decoupled BLD for subspace narrowing with coupled BLD for refinement improves accuracy while maintaining computational costs (see Appendix F.1.1).

- **Dataset Composition:** Models trained on the diverse Distillation Mix outperformed those trained on the limited-domain Project Gutenberg, but Gutenberg-trained models still retained ~93% of performance, showcasing Puzzle's robustness (see Appendix F.1.2).

- **Training Dataset Size:** BLD achieves strong performance even with smaller token budgets, with diminishing returns beyond 0.5B tokens (see Appendix F.1.3).

- **Block Scoring Metrics:** KL divergence scoring outperformed LM loss and task-specific downstream scoring, demonstrating better balance between generality and accuracy, although task-specific scoring provides an advantage for customized target tasks (see Appendix F.1.4).

- **Reduced Search Space:** Constraining the search space to no-op alternatives simplifies optimization and eliminates BLD but results in lower accuracy (75.4 vs. 78.39 MMLU), highlighting the value of diverse block replacements for optimal performance (see Appendix F.1.5).

#### F.1.1. COMBINING COUPLED BLD AND DECOUPLED BLD

We investigate the effects of coupling in BLD (see Section 3.1) on Puzzle derivatives of Llama-3.1-8B-Instruct, which has 32 layers. For each layer, the search space we consider contains $|\mathcal{A}_i| = 6$ variants of the attention subblock and $|\mathcal{F}_i| = 12$ variants of the FFN subblock. With coupled BLD, this would amount to training $6 \cdot 12 \cdot 32 = 2304$ blocks. Decoupled BLD reduces the training requirements to only $(4 + 10) \cdot 32 = 448$ subblocks, which is considerably more resource-efficient. Note that besides moving from multiplicative composition to additive composition, with decoupled BLD we also do not need to train the no-op and parent variants in $\mathcal{A}_i$ and $\mathcal{F}_i$.

We propose a two-stage technique to reap the benefits of coupled BLD while still keeping the computational cost reasonable. First, we run the full Puzzle framework with decoupled BLD. Then, we analyze the architectures produced by the search algorithm, and identify the most prominent choices of subblock variants. We shrink the search space to include only these choices, then run the full Puzzle framework with coupled BLD on the reduced search space. In our case, we were able to shrink $|\mathcal{A}_i|$ from 6 to 4, and $|\mathcal{F}_i|$ from 12 to 3, resulting in a total number of $4 \cdot 3 \cdot 32 = 384$ blocks to train in coupled BLD, which is similar to the number we trained in decoupled BLD for the larger search space. This approach produced a higher-accuracy architecture. Note, however, that the one created with decoupled BLD was already significantly above the efficient frontier. See results in Table 8. Both models underwent short GKD uptraining.

*Table 7.* Performance comparison of the parent (Llama-3.3-70B-Instruct) and child (Nemotron-49B-Base) on a subset of the RULER benchmark. 'Accuracy Preserved' is (child / parent)×100.

| Context | single_needle_1 | single_needle_2 | single_needle_3 | multi_needle_1 | multi_needle_2 | multi_needle_3 | multi_value | multi_query | variable_tracking_1_4 | common_words_extraction | freq_words_extraction | qa_squad | qa_hotpotqa | Average | Accuracy Preserved (%) |
|---|---|---|---|---|---|---|---|---|---|---|---|---|---|---|---|
| **Parent: Llama-3.3-70B-Instruct** | | | | | | | | | | | | | | | |
| 4K | 100 | 100 | 100 | 100 | 100 | 100 | 100 | 100 | 100 | 95 | 91 | 72 | 96.77 | 96.77 | - |
| 8K | 100 | 100 | 100 | 100 | 100 | 100 | 100 | 100 | 99.8 | 96 | 87 | 71 | 96.46 | 96.46 | - |
| 16K | 100 | 100 | 100 | 100 | 100 | 100 | 100 | 100 | 100 | 95 | 86 | 67 | 95.98 | 95.98 | - |
| 32K | 100 | 100 | 100 | 100 | 100 | 99 | 98.75 | 100 | 95 | 92.33 | 83 | 63 | 94.70 | 94.70 | - |
| 64K | 100 | 100 | 100 | 100 | 97 | 93 | 98.25 | 100 | 43.9 | 92.67 | 75 | 56 | 88.91 | 88.91 | - |
| 128K | 37 | 100 | 100 | 90 | 0 | 1 | 98.25 | 95.75 | 0.4 | 3.5 | 78.33 | 41 | 34 | 52.25 | 52.25 | - |
| **Child: Nemotron-49B-Base** | | | | | | | | | | | | | | | |
| 4K | 100 | 100 | 100 | 100 | 100 | 100 | 100 | 100 | 100 | 95.9 | 99.33 | 93 | 78 | 97.40 | **100.65** |
| 8K | 100 | 100 | 100 | 100 | 100 | 100 | 100 | 100 | 100 | 92.4 | 98.33 | 89 | 76 | 96.59 | **100.13** |
| 16K | 100 | 100 | 100 | 100 | 99 | 100 | 99.75 | 100 | 100 | 92.4 | 98 | 89 | 71 | 96.09 | **100.11** |
| 32K | 100 | 100 | 100 | 100 | 99 | 99 | 99.5 | 100 | 81.4 | 95 | 85 | 67 | 94.3 | 94.30 | **99.58** |
| 64K | 100 | 100 | 100 | 100 | 92 | 92 | 99.75 | 100 | 26.9 | 92.67 | 73 | 60 | 87.39 | 87.39 | **98.29** |
| 128K | 45 | 99 | 99 | 89 | 0 | 0 | 88 | 86.5 | 1.6 | 1.3 | 56.67 | 44 | 35 | 49.62 | **94.97** |

*Table 8.* The effect of coupled BLD vs decoupled BLD on high-throughput child derivatives of Llama-3.1-8B-Instruct. We found a relevant subspace of the search space using a decoupled BLD Puzzle, then trained coupled BLD on this subspace and ran a separate Puzzle, leading to additional improvement. Throughput is estimated via the sum of measured block runtimes on a single NVIDIA RTX 4090 GPU. Accuracy = (MT-Bench ×10 + MMLU) / 2.

| Model | Throughput* | Accuracy |
|---|---|---|
| Puzzle with Coupled BLD | 5856 | 73.98 |
| Puzzle with Decouple BLD | 5834 | 73.10 |
| Llama-3.2-3B-Instruct | 5737 | 70.36 |

### F.1.2. IMPACT OF DATASET COMPOSITION ON PUZZLE-DERIVED MODELS

We evaluate the robustness of the Puzzle framework to training data composition by comparing performances up through the BLD stage (prior to GKD). For this analysis, we contrast two datasets: our domain-diverse Distillation Mix (described in Section 3) and the English subset of Project Gutenberg (Project Gutenberg). The latter, a dataset predominantly comprising literary works, which lacks diverse coverage of technical, conversational and STEM-specific content, making it an interesting test case for framework robustness.

As shown in Table 9, models derived using Project Gutenberg data (for training, pruning and block scoring) demonstrate strong performance despite the dataset's limitations. On general benchmarks like MT-Bench and MMLU, the Gutenberg-trained model achieves 92.7% and 95.5% of the performance obtained with Distillation Mix, respectively. Even on STEM categories within MMLU, where the training data's limitations are most relevant, the model maintains 91.7% of the performance (64.5 vs 70.35).

These results demonstrate that the Puzzle framework can effectively transfer knowledge from the parent model even when the training data provides limited coverage of specific domains. This robustness is particularly noteworthy given that no GKD uptraining was performed, suggesting that our BLD approach effectively preserves model capabilities across domains regardless of the training data composition.

*Table 9.* Benchmark results on Llama-3.1-70B-Instruct derivatives obtained from Puzzle without uptraining applied with different datasets.

| Model | MT-Bench | MMLU | MMLU-STEM |
|---|---|---|---|
| Gutenberg-Trained | 7.98 | 74.84 | 64.5 |
| DistillationMix-Trained | 8.61 | 78.39 | 70.35 |

### F.1.3. IMPACT OF BLOCKWISE LOCAL DISTILLATION TRAINING DATASET SIZE

To evaluate the efficiency of BLD under varying training dataset sizes, we conducted experiments using different token budgets. Specifically, we trained the same set of block variants using 0.25B, 0.5B, and 1.0B tokens from the *Distillation Mix* dataset (see Section 3). After completing the BLD stage for each token budget, we used the MIP optimization stage to generate optimized architectures under similar constraints to those used to produce Nemotron-51B, and evaluated their performance on downstream tasks.

In Table 10 we present the performance of models trained with different BLD token budgets. Notably, while all models achieved comparable MMLU scores, the MT-Bench results indicate a more pronounced performance improvement as the token budget increases, particularly in multi-turn conversational tasks. However, the improvements diminish as the token budget grows (e.g., the boost from 0.25B to 0.5B tokens is larger than that from 0.5B to 1.0B), suggesting that BLD facilitates rapid recovery of the parent model's performance even with moderate training budgets. These findings imply that longer BLD training can yield better block libraries but with diminishing returns at larger scales.

### F.1.4. IMPACT OF DIFFERENT BLOCK SCORING METRICS

In Section 4.2, we presented three possible metrics for replace-1-block scores: downstream accuracy, LM loss and KL divergence. We investigate the effects of using different replace-1-block scores when applying the Puzzle framework to Llama-3.1-8B-Instruct with a large search space

*Table 10.* Performance comparison of Puzzle-optimized architectures trained with varying BLD token budgets. Metrics include MT-Bench and MMLU scores.

| BLD Token Budget | MT-Bench | MMLU |
|---|---|---|
| 1.0B Tokens | 8.98 | 78.54 |
| 0.5B Tokens | 8.86 | 78.44 |
| 0.25B Tokens | 8.51 | 78.27 |

$(|\mathcal{A}_i| = 6, |\mathcal{F}_i| = 12)$. We compare the use of the model's loss on a validation set (LM loss in our case), a common choice in earlier decomposed NAS methods used in CV, with our proposed method based on KL divergence. LM loss aims to capture the general quality degradation induced by changing a specific parent block, while KL divergence aims to capture the distance from the parent model induced by this change. As illustrated in Figure 7, KL divergence scoring results in better Puzzle architecture choices than scoring with LM loss. All models were constructed using decoupled BLD and underwent short GKD uptraining.

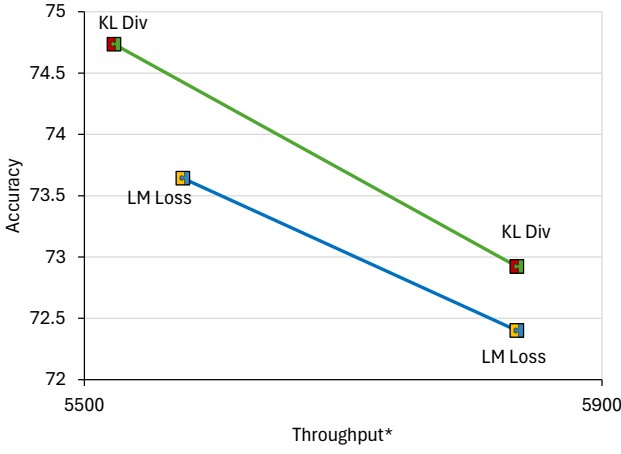

*Figure 7.* Accuracy vs. Throughput performance of Llama-3.1-8B-Instruct child derivatives, some constructed using LM loss as replace-1-block scores, and some constructed using KL divergence as replace-1-block scores. LM loss aims to capture the general quality degradation induced by changing a specific parent block, while KL divergence aims to capture the distance from the parent model induced by this change. KL divergence results in better architecture choices. Accuracy = (MT-Bench ×10 + MMLU) / 2. Throughput is estimated via the sum of measured block runtimes on a single NVIDIA RTX 4090 GPU.

Next, we explore the use of downstream accuracy as replace-1-block scores to customize the algorithm's block selection for specific target tasks. Our hypothesis is that, within the same budget, different architectures may excel at different tasks because distinct capabilities—such as reasoning,

world knowledge, or conversational abilities—are likely concentrated in different parts of the model. Due to the computational expense of calculating downstream accuracy, we use the reduced search space from Appendix F.1.1 to keep the experiment feasible. To evaluate downstream accuracy, we use the MMLU benchmark, splitting its 57 tasks into two nearly equal-sized sets stratified by the MMLU categories {STEM, Social Sciences, Humanities, Other}. These subsets are referred to as *Half-MMLU*, with one serving as a "train" set for block quality scoring and the other as a "test" set for evaluation. Table 11 shows that, even with the same library of trained blocks, the block selection process can be customized to construct architectures optimized for specific target tasks. We note that this customization is not without cost: the architecture constructed using Half-MMLU scores achieves the best accuracy on the target task, but its MT-Bench accuracy of 7.57 is lower than the 8.06 MT-Bench accuracy of the architecture constructed according to the more general KL divergence scores. Both models were constructed using decoupled BLD and underwent short GKD uptraining.

*Table 11.* The effect of task-oriented block scoring on high-throughput child derivatives of Llama-3.1-8B-Instruct. We split the tasks in MMLU into two equal-sized sets and use one of them for block quality scoring and the other for evaluation, showing that even with the same library of trained blocks, block selection can be customized to build architectures that fit a desired target task. Throughput is estimated via the sum of measured block runtimes on a single NVIDIA RTX 4090 GPU.

| Model | Throughput* | Half-MMLU Accuracy (Test Set) |
|---|---|---|
| Puzzle: scored with Half-MMLU accuracy (train set) | 5818 | 66.24 |
| Puzzle: scored with KL divergence | 5834 | 64.94 |
| Llama-3.2-3B-Instruct | 5737 | 60.06 |

### F.1.5. EFFECTS OF LIMITED SEARCH SPACE DIVERSITY

To explore the impact of reducing the search space complexity on the Puzzle framework, we constrained alternative child blocks to only allow replacing parent model subblocks with no-op operations. This eliminates the need for BLD as no additional block variants that require training are considered, further reducing the computational costs.

The resulting architecture, optimized using the same MIP approach but limited to no-ops, was evaluated against pre-uptraining Nemotron-51B, which was derived using the same Puzzle pipeline but with a more diverse block variants. As shown in Table 12, the no-op-only model retained high throughput but exhibited a noticeable drop in MMLU accuracy compared to Nemotron-51B (75.4 vs. 78.39).

These results illustrate the flexibility of the Puzzle framework in balancing resource constraints and model performance. Although limiting the search space simplifies the optimization process and reduces training costs even further,

it sacrifices the fine-grained architectural customization enabled by a broader range of block alternatives. This demonstrates that a more diverse search space leads to architectures that achieve better accuracy.

*Table 12.* Comparison of pre-uptraining Nemotron-51B (derived using the full search space) and a no-op-only variant.

| Model | MMLU | Throughput (tokens/sec) |
|---|---|---|
| Puzzle (No-op only) | 75.40 | 5604.18 |
| Puzzle (Full search space) | 78.39 | 5500.25 |

## F.2. Search Algorithm Ablation Studies

In Appendices F.2.1 to F.2.3, we analyze the MIP optimization process and alternative approaches within the Puzzle framework. These studies focus on how throughput constraints and scoring strategies influence architecture selection and model performance. Our key findings are:

- **MIP Optimization and Throughput Constraints:** The MIP solver adapts architectures to meet various throughput targets, revealing nuanced trade-offs between global and local efficiency. Notably, FFN components are preserved even under strict constraints, highlighting their critical role in maintaining model accuracy (see Appendix F.2.1).

- **Greedy Algorithm Alternative:** A budget-constrained greedy algorithm, while simpler, resulted in significantly lower accuracy (70.74% MMLU) compared to MIP-based optimization (78.39%). This underscores the importance of global optimization for achieving superior accuracy-efficiency trade-offs (see Appendix F.2.2).

- **Data-free Scoring:** Maximizing parameter count as a heuristic produced architectures with sharp performance declines (23.12% MMLU), emphasizing the necessity of data-driven scoring mechanisms for effective architecture optimization (see Appendix F.2.3).

- **Random Architecture Baselines:** Nemotron-51B outperforms both fully random and random-from-library architectures under the same training and speed constraints, underscoring the role of the MIP solver in selecting effective architectural compositions (see Appendix F.2.4).

### F.2.1. MIP SOLUTION ANALYSIS UNDER VARYING THROUGHPUT CONSTRAINTS

Figure 8 provides an interesting window into how our MIP solver adapts model architecture to different throughput requirements. Each row in the heatmaps represents a distinct architecture optimized for a specific throughput target, with darker colors indicating higher computational cost relative to the parent model. The architecture of Nemotron-51B, corresponding to a throughput target of 5500 tokens per second, is marked in green. Several intriguing patterns are evident:

- **Counter-intuitive local optimizations:** While stricter throughput constraints generally lead to faster blocks, we observe surprising inversions of this trend. For example, in layers 1-4, the MIP sometimes chooses computationally heavier blocks for higher throughput targets. This counter-intuitive choice suggests that local slowdowns can enable better global optimization, with other layers compensating to meet the overall throughput constraint.

- **Asymmetric treatment of components:** The MIP treats attention and FFN components quite differently. While attention mechanisms are completely eliminated in some layers even under lenient throughput constraints, FFN components are never entirely skipped. This suggests that FFN layers might play a more fundamental role in maintaining model capabilities, while attention mechanisms offer more flexibility for optimization.

- **Architectural phases:** The heatmap reveals distinct "phases" across layer depths. Shallow layers (0-15) show high variability in both attention and FFN, middle layers (16-42) maintain more consistent computation, and deeper layers (43-80) show different patterns for attention versus FFN optimization. This suggests different layers serve distinct roles in the network's information processing.

- **Throughput-dependent transitions:** The solution patterns show clear transitions as throughput requirements increase, but these transitions are not uniform across layers. Some layers maintain consistent computation across different throughput targets while others show sharp transitions, indicating varying sensitivity to throughput constraints.

These patterns demonstrate the sophisticated optimization strategies discovered by the MIP solver, revealing that optimal architectures often require nuanced tradeoffs between local and global efficiency. The preservation of FFN computation even under strict constraints provides empirical evidence for the relative importance of different architectural components in such transformer models.

### F.2.2. GREEDY SEARCH ALGORITHM ALTERNATIVE

To further evaluate the impact of the MIP optimization approach, we implemented a budget-constrained greedy search

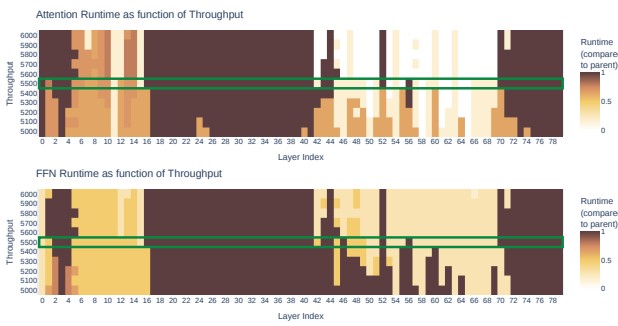

*Figure 8.* Heatmaps showing how attention and FFN runtime patterns vary with throughput constraints across model layers. Dark colors indicate higher computational cost relative to the parent model. Each row represents an architecture optimized for a specific throughput target, with Nemotron-51B's configuration (5500 tokens/sec) marked in green.

algorithm as an alternative. This algorithm prioritizes simplicity by using a heuristic-based layer-wise selection process, contrasting with the global optimization provided by MIP.

The greedy algorithm operates as follows:

- At initialization, the runtime and memory budgets which are derived from the required constraints, are split equally across layers.

- Layer scoring: Each layer is assigned a score based on a heuristic metric designed to estimate how easily it can be replaced with minimal performance degradation. In our implementation, this metric was the mean replace-1-block KL divergence score across all block variants for the layer. Layers with lower average scores were deemed easier to optimize.

- Sequential replacement: Layers are processed in ascending order of their scores. For each layer, the algorithm selects the block variant with the lowest replace-1-block KL divergence score that satisfies the layer's runtime and memory budget.

- Constraint adjustment: After selecting a block for the current layer, the remaining runtime and memory savings are added to the next layer's budget. This allows for a more dynamic algorithm that allocates more resources for layers that are harder to replace.

Table 13 compares the performance of the greedy algorithm with the MIP-derived pre-uptraining Nemotron-51B. The results show that the greedy algorithm leads to a significant drop in accuracy, highlighting the importance of global optimization. Specifically, the model derived using the greedy

algorithm achieves a substantially lower MMLU accuracy of 70.74%, compared to 78.39% for the MIP-derived model, despite both architectures meeting the same throughput constraint.

*Table 13.* Comparison of the budget-constrained greedy algorithm and MIP as search algorithms for Puzzle. Results are shown for pre-uptraining Nemotron-51B under identical throughput constraints.

| Optimization Method | MMLU | Throughput (tokens/sec) |
|---|---|---|
| Greedy Algorithm | 70.74 | 5500.30 |
| MIP | 78.39 | 5500.25 |

These findings underscore the critical role of global optimization in the Puzzle framework. By considering jointly optimizing block selection, MIP achieves a significantly better balance between efficiency and accuracy, making it indispensable for extracting the full potential of the Puzzle framework.

### F.2.3. IMPORTANCE OF DATA-DRIVEN QUALITY ESTIMATION IN ARCHITECTURE SCORING

To further understand the importance of quality estimation in search space optimization, we conducted an experiment using a simple heuristic: scoring blocks based on their parameter count. While not a data-driven approach, parameter count serves as a straightforward metric often associated with improved performance in LLMs. This experiment aimed at assessing whether such a basic metric could provide sufficient guidance in the search space, shedding light on the role of more nuanced scoring mechanisms.

Under this method, the search algorithm is simplified to selecting the block variant with the largest number of parameters that satisfied throughput and memory constraints. This resulted in an architecture composed uniformly of high-parameter blocks across all layers, without considering layer-specific block quality and representational needs.

As shown in Table 14, maximizing parameter count leads to a sharp drop in MMLU accuracy compared to the architecture derived using the full Puzzle framework with MIP optimization. Despite having similar throughput, the simplistic parameter-maximization approach fails to achieve competitive results, underscoring the necessity of quality-aware block scoring, and different layers require different computational budgets for optimal performance.

### F.2.4. EVALUATING RANDOM ARCHITECTURES AS SEARCH BASELINES

To further validate the effectiveness of our search algorithm, we evaluated several baselines where block selection was performed randomly. All models were trained with 10B tokens.

*Table 14.* Comparison of maximizing parameter count with Puzzle's MIP-based optimization as search algorithms. Results are shown for pre-uptraining Nemotron-51B under identical throughput constraints.

| Optimization Method | MMLU | Throughput (tokens/sec) |
|---|---|---|
| Maximizing Parameters | 23.12 | 5727.08 |
| pre-uptraining Nemotron-51B | 78.39 | 5500.25 |

The first baseline ("Random-from-block-library") randomly samples block variants from the full block library while satisfying the same throughput constraint as Nemotron-51B, ignoring their block scores. The second ("Fully Random") uses a completely random architecture that is not constrained to trained block variants, but adheres to the same speed constraints. Finally, we added a third baseline—*Parent-Randomized*—which evaluates Llama-3.1-70B with randomized weights but no architectural modifications.

As shown in Table 15, Puzzle significantly outperforms all baselines. Notably, the random-from-library baseline achieves only 86.6% of Nemotron-51B's performance, despite being constructed from the same trained blocks. This experiment further emphasizes the role of MIP in selecting high-quality architectural compositions from the library.

*Table 15.* Comparison of Nemotron-51B with randomly constructed architectures, all trained for 10B tokens. The random-from-library variant uses trained blocks, while the fully random variant ignores the block library. The Parent-Randomized model uses Llama-70B's architecture with random weights.

| Model | MMLU | MT-Bench | Avg. Accuracy | Relative to Llama-70B |
|---|---|---|---|---|
| Nemotron-51B (10B tokens) | 79.7 | 8.89 | 84.30 | 98.61% |
| Random-from-block-library | 66.02 | 8.20 | 74.01 | 86.58% |
| Fully Random | 23.13 | 0.89 | 16.02 | 18.73% |
| Parent-Randomized | 23.42 | 0.95 | 16.46 | 19.25% |
| Llama-3.1-70B | 81.66 | 8.93 | 85.48 | 100% |

## F.3. Global Knowledge Distillation Uptraining

We assess the significance of the final GKD uptraining phase, in enhancing the accuracy of child models derived from Llama-3.1-70B-Instruct and Llama-3.1-8B-Instruct. The results presented in Table 16 demonstrate that this stage contributes to improvements in both the MMLU and MT-Bench benchmark scores.

*Table 16.* Impact of global knowledge distillation uptraining on MMLU and MT-Bench benchmark scores for child models derived from Llama-3.1-70B-Instruct and Llama-3.1-8B-Instruct.

| Model Name | GKD Uptraining | MMLU | MT-Bench | Average |
|---|---|---|---|---|
| Llama-3.1-70B-Instruct (parent) | - | 81.66 | 8.93 | 85.48 |
| Nemotron-51B-Instruct (child) | ✗ | 78.39 | 8.67 | 82.55 |
| | ✓ | 80.20 | 8.99 | 85.10 |
| Llama-3.1-8B-Instruct (parent) | - | 69.40 | 8.34 | 76.40 |
| Child derivative of Llama-3.1-8B-Instruct (child) | ✗ | 65.25 | 7.29 | 69.06 |
| | ✓ | 65.46 | 8.25 | 73.98 |

### F.3.1. LOSS COMPOSITION ABLATION STUDIES FOR UPTRAINING

Recall from Section 4.3 that our final GKD objective comprises cosine similarity and KL divergence losses:

$$\mathcal{L}_{\text{cosine}} = \sum_{l=1}^{L} \left( 1 - \frac{\mathbf{h}_c^l \cdot \mathbf{h}_p^l}{\|\mathbf{h}_c^l\| \|\mathbf{h}_p^l\|} \right), \qquad (2)$$

$$\mathcal{L}_{\text{KLD}} = \sum_{i=1}^{N} p_i \log\left( \frac{p_i}{q_i} \right), \qquad (3)$$

In this subsection, we provide a detailed ablation of different loss combinations, including the *language modeling (LM) loss*:

$$\mathcal{L}_{\text{LM}} = -\sum_{i=1}^{N} y_i \log(\hat{y}_i), \qquad (4)$$

where $y_i$ is the ground-truth label for the $i$-th token and $\hat{y}_i$ its predicted probability. We paired $\mathcal{L}_{\text{LM}}$ with cosine and KL divergence losses in various configurations to see which composition best recovers the parent model's accuracy.

**Experimental Setup:** We used Nemotron-51B (the immediate child after BLD) and trained with $\sim 5$B tokens for each combination, measuring accuracy on MMLU and MT-Bench. Table 17 shows that adding $\mathcal{L}_{\text{LM}}$ often harms downstream performance, consistent with Muralidharan et al. (2024). In contrast, combining $\mathcal{L}_{\text{cosine}}$ and $\mathcal{L}_{\text{KLD}}$ yields the best results. We thus adopt Equation (1):

$$\mathcal{L}_{\text{GKD}} = \mathcal{L}_{\text{cosine}} + \mathcal{L}_{\text{KLD}}$$

as our final uptraining loss, ultimately running it over 45B tokens to produce Nemotron-51B-Instruct.

*Table 17.* Ablation study for different combinations of LM loss, block (hidden activations) loss, and logits KLD loss. All models (Nemotron-51, derived from Llama-3.1-70B-Instruct) were trained for $\sim 5B$ tokens. First row did not undergo uptraining. Adjacent rows with the same color differ only in the $\mathcal{L}_{\text{LM}}$ component. *During the KD process for this combination, the validation $\mathcal{L}_{\text{KLD}}$ consistently increased. †Trained for 45B tokens using $\mathcal{L}_{\text{GKD}}$ defined in Equation (1).

| $\mathcal{L}_{\text{LM}}$ (4) | $\mathcal{L}_{\text{cosine}}$ (2) | $\mathcal{L}_{\text{KLD}}$ (3) | MMLU | MT-Bench | Average | Validation $\mathcal{L}_{\text{KLD}}$ |
|---|---|---|---|---|---|---|
| ✗ | ✗ | ✗ | 78.39 | 8.67 | 82.55 | 0.19 |
| ✓ | ✗ | ✗ | 78.55 | 7.71 | 77.83 | 0.31* |
| ✓ | ✗ | ✓ | 79.26 | 8.85 | 83.88 | 0.14 |
| ✗ | ✗ | ✓ | 79.33 | 8.68 | 83.07 | **0.10** |
| ✓ | ✓ | ✗ | 79.04 | 7.80 | 78.52 | 0.30* |
| ✗ | ✓ | ✗ | 79.40 | 8.74 | 83.40 | 0.16 |
| ✓ | ✓ | ✓ | 79.45 | 8.66 | 83.03 | 0.14 |
| ✗ | ✓ | ✓ | **79.61** | **8.87** | **84.16** | 0.11 |
| Llama-3.1-70B-Instruct (parent) | | | 81.66 | 8.93 | 85.48 | 0.00 |
| Nemotron-51B-Instruct (child)† | | | 80.20 | 8.99 | 85.10 | 0.08 |

We hypothesize that $\mathcal{L}_{\text{LM}}$ may cause overfitting to the uptraining dataset, thus reducing the child's ability to mirror the parent's distribution across downstream tasks. Removing this term mitigates open-weights, closed-data mismatches, leading to better knowledge transfer.

## F.4. Comparison with Related Work

There are many different techniques to prune or compress models, most of which could be complementary to Puzzle. Some pruning methods, for example, could be used to construct alternative blocks for Puzzle's block library. Nevertheless, in this section we compare several established methods directly against the basic Puzzle framework, without integrating them.

We compare Puzzle against structured sparsity (Wanda (Sun et al., 2024)) and low-rank approximation (similar to (Khodak et al., 2021) with subsequent distillation) applied to Llama-3.1-70B under similar throughput constraints. Wanda applied 2:4 structured sparsity without additional training, while the low-rank method used factorized layers followed by distillation. Nemotron-51B significantly outperformed both methods, achieving 99.49% of the parent's average accuracy (MMLU and MT-Bench), compared to 92.23% for Wanda and 88.96% for low-rank approximation (see Table 18). Subsequent distillation post-pruning with Wanda slightly improved MMLU (73.69) without impacting MT-Bench. Moreover, since both structured sparsity and low-rank approximations represent subsets of Puzzle's broader search space, integrating these approaches into Puzzle could further enhance performance.

*Table 18.* Comparison of Puzzle, Wanda (structured sparsity), and low-rank approximation methods on Llama-3.1-70B derivatives under similar throughput constraints.

| Model | MMLU | MT-Bench | Average Accuracy | Accuracy Preserved (%) |
|---|---|---|---|---|
| Nemotron-51B | 80.20 | 8.99 | 85.05 | 99.49% |
| Wanda (Sun et al., 2024) | 72.99 | 8.39 | 78.44 | 92.23% |
| Low-rank | 72.87 | 8.01 | 76.05 | 88.96% |
| Llama-3.1-70B (Parent) | 81.66 | 8.93 | 85.48 | 100% |

Other methods such as Minitron (Muralidharan et al., 2024), ShortGPT (Men et al., 2024), and SlimGPT (Ling et al., 2024) share conceptual similarities with Puzzle, but each represents a constrained subset of Puzzle's broader optimization space. For the most part, Minitron restricts modifications to homogeneous block replacements across all layers, ShortGPT focuses exclusively on redundant layer removal, and SlimGPT employs incremental pruning ratios via a fixed heuristic. Puzzle generalizes and extends these approaches, allowing for heterogeneous, layer-specific modifications, diverse block alternatives including no-op layers, and customizable pruning ratios optimized globally through MIP-based optimization.

*Table 19.* Full performance comparison of the parent (Llama-3.1-70B-Instruct) and child (Nemotron-51B) on a subset of the RULER benchmark across all context lengths. 'Accuracy Preserved' is (child / parent)×100. Benchmark names refer to the implementation/settings in the official RULER repository. *Varies depending on context length.

| Context Length | qa_hotpotqa | qa_squad | common_words_extraction* | variable_tracking_1_4 | variable_tracking_2_2 | freq_words_extraction_2 | freq_words_extraction_3_5 | Average | Accuracy Preserved (%) |
|---|---|---|---|---|---|---|---|---|---|
| **Parent: Llama-3.1-70B-Instruct** | | | | | | | | | |
| 1K | N/A | N/A | 100.00 | 100.00 | 100.00 | 99.40 | 99.67 | 99.81 | - |
| 2K | N/A | 88.40 | 100.00 | 100.00 | 100.00 | 99.53 | 99.87 | 97.97 | - |
| 4K | 67.60 | 87.40 | 100.00 | 100.00 | 99.87 | 99.00 | 99.80 | 93.38 | - |
| 8K | 67.80 | 83.80 | 99.96 | 100.00 | 99.40 | 97.87 | 99.93 | 92.68 | - |
| 16K | 63.20 | 82.00 | 98.86 | 100.00 | 96.87 | 96.67 | 99.93 | 91.08 | - |
| 32K | 61.60 | 77.20 | 93.48 | 100.00 | 97.93 | 95.53 | 100.00 | 89.39 | - |
| 64K | 55.40 | 72.60 | 26.16 | 99.96 | 97.93 | 94.53 | 99.87 | 78.06 | - |
| 128K | 33.65 | 49.04 | 2.37 | 56.85 | 36.33 | 78.61 | 85.71 | 48.94 | - |
| **Child: Nemotron-51B** | | | | | | | | | |
| 1K | N/A | N/A | 99.98 | 100.00 | 100.00 | 99.40 | 99.53 | 99.78 | 99.90 |
| 2K | N/A | 86.20 | 99.86 | 99.96 | 99.67 | 98.40 | 99.80 | 97.32 | 99.34 |
| 4K | 63.40 | 85.00 | 99.92 | 100.00 | 98.93 | 97.73 | 99.87 | 92.12 | 98.65 |
| 8K | 58.20 | 80.80 | 99.34 | 100.00 | 99.60 | 96.67 | 99.80 | 90.63 | 97.79 |
| 16K | 53.40 | 75.60 | 93.50 | 99.72 | 96.80 | 94.73 | 99.80 | 87.65 | 96.23 |
| 32K | 45.60 | 70.60 | 51.92 | 98.28 | 93.67 | 90.27 | 99.47 | 78.54 | 87.86 |
| 64K | 7.40 | 15.20 | 2.28 | 3.48 | 7.87 | 36.93 | 8.67 | 11.60 | 14.86 |
| 128K | 3.80 | 3.20 | 0.10 | 0.00 | 0.00 | 2.07 | 0.00 | 1.31 | 2.67 |

