# OpenReview forum: "Puzzle: Distillation-Based NAS for Inference-Optimized LLMs"
_ICML.cc/2025/Conference — ICML 2025 poster_

### Official Review · Reviewer_Fna8 · 2025-03-09

**Overall Recommendation:** 5

**Summary:**

The paper introduces Puzzle, a distillation-based NAS approach for extracting inference-optimized LLMs for existing trained models such as Llama. The authors first introduce the search space for their NAS-based optimizations - including the different attention and FFN subblocks to use, followed by the number of combinations, and introduce a decoupled block distillation algorithm to reduce the number of total combinations to explore for search. Following this, they define their overall algorithm for Puzzle:
- Search constraints focused on real-time memory and throughput on H100 GPUs
- Scoring method for each subblock
- Mixed Integer Programming for selecting the best solution with their constraints
- Additional global training to improve overall model quality

For results, the authors present the 51B model, which is distilled from Llama 70B, including evaluations on evaluation benchmarks, blind test comparisons, training a 49B model with similar memory requirements but with longer context length support, and a final smaller model distilled from the 8b model.

-----------

### After Rebuttal
Based on the provided rebuttal and the other reviews / addressed comments, I've decided to retain my score. All concerns have been addressed.

**Claims And Evidence:**

Yes, the claims made in the submission have sufficient evidence for each aspect. Just highlighting some of them below:

- The authors present a series of ablations for aspects of the algorithm, for example, the scoring method for each sub-block (using KL vs cross entropy vs accuracy on downstream tasks), which datasets they used for their BLD, and how long to train each sub-block for during exploration. The ablations are empirically supported by results from reasonable downstream evaluations, measuring throughput, etc.
- For the mixed integer programming algorithm, they add a diversity constraint, and show graphs representing how different blocks have different sub-blocks selected (see Fig. 6 for example).

**Essential References Not Discussed:**

N/A

**Experimental Designs Or Analyses:**

Yes, I did check the design and associated analysis. Here are some follow up questions:

- For the throughput comparison of table 2, can the authors clarify what batch sizes they used for the numbers?
- For the same table, when using TP=1 for the 128/1024 scenario (entry 2) - the slow throughput for Llama-70B almost seems to be an issue of hitting the memory + optimization wall very quickly with an H100 80GB instance rather than the model not being "efficient" here. Can the authors comment on this? If so, comparing with say TP=2 might have been a better comparison point vs TP=4 here?
- For Figure 5, can the authors clarify if they used MMLU or MMLU Chat for the accuracy computations?

**Methods And Evaluation Criteria:**

Yes, most of the methods and the equivalent evaluation criteria make sense for the problem the authors are solving.

**Other Comments Or Suggestions:**

N/A

**Other Strengths And Weaknesses:**

**Strengths**
- The paper does thorough ablations of each decision choice in the creation of the algorithm
- For the NAS objective to minimize, the authors consider real-world deployments over theoretical measures such as parameter count or FLOPs
- The approach identifies potential algorithm pieces that may introduce extra complexity (such as the large search space) and focuses on solving those through feedback from ablations.


**Weakness**
The paper discusses an approach to find smaller networks from an existing trained LLM using the suggested NAS-based approach for efficiency. This is similar to other work done previously to find smaller networks such as Minitron [1], but also other pruning approaches such as ShortGPT [2] and SlimGPT [3]. However, none of these approaches are explicitly compared in the paper, beyond trying to recover the accuracy of the original trained model.

[1] Minitron: https://arxiv.org/abs/2408.11796
[2] ShortGPT: https://arxiv.org/abs/2403.03853
[3] SlimGPT: https://arxiv.org/abs/2412.18110

**Questions For Authors:**

N/A

**Relation To Broader Scientific Literature:**

The paper presents a NAS+distialltion based approach to get best in class LLMs for training. Given the complexity of using NAS for LLMs, the authors present new approaches to reduce complexity for NAS, especially dealing with sub-block optimizations that are initially greedy (optimizing for that block w/ inputs only from previous blocks). This is similar to approaches like LayerNAS [1] which have been previously explored for CV architectures.


[1] LayerNAS: https://arxiv.org/abs/2304.11517

**Theoretical Claims:**

N/A

---

> ### Author Rebuttal · Authors · 2025-03-31
>
> Thank you for your positive feedback!,
>
> *"For the throughput comparison of table 2, can the authors clarify what batch sizes they used for the numbers?"*
>
> This is a good question. For every model and hardware setting we selected the optimal batch size to get the best throughput per GPU. This was done automatically by the inference engine, selecting for each run the optimal batch size to use. For example, Puzzle-51B's optimal batch size for TP=1 was 256, and for Llama-3.1-70B at TP=4, it was 384. We will make sure to note this in our revision. Do you think it is beneficial to include the full list of batch sizes as well?
>
> *"For the same table, when using TP=1 for the 128/1024 scenario (entry 2) - the slow throughput for Llama-70B almost seems to be an issue of hitting the memory + optimization wall very quickly with an H100 80GB instance rather than the model not being "efficient" here. Can the authors comment on this? If so, comparing with say TP=2 might have been a better comparison point vs TP=4 here?"*
>
> You are right about TP=1. As described in the caption, for each model the optimal TP was chosen for speed measurements, which is why we selected TP=4 and not TP=2 for Llama-70B: TP=4 is more flattering to the parent's throughput for this scenario.
> We also included the speed measurements on TP=1 to also present a "fair" comparison to Puzzle-51B for a single GPU setting. We will clarify this point explicitly in the revision.
>
> *"For Figure 5, can the authors clarify if they used MMLU or MMLU Chat for the accuracy computations?"*
>
> We used MMLU (and not MMLU Chat).
>
> *"This is similar to other work done previously to find smaller networks such as Minitron [1], but also other pruning approaches such as ShortGPT [2] and SlimGPT [3]. However, none of these approaches are explicitly compared in the paper"*
>
> You make a good point, and we are working on including these relevant comparisons in the revision.
> In short, some of these methods share similar aspects to Puzzle, but their solutions remain a subset of Puzzle's search space. For example, Minitron only considers homogeneous solutions (i.e., any modified block is applied across all layers), whereas Puzzle allows for heterogeneous, layer-specific configurations.
> ShortGPT considers layer removal (similar to how Puzzle allows "no-op" layers), using a cosine similarity score to identify redundant layers.
> SlimGPT introduces an extension of the "Optimal Brain Surgeon" method, called "Batched Greedy Pruning", that could also be used by Puzzle to prune individual blocks. SlimGPT sets an "Incremental Pruning Ratio" heuristic that follows a fixed logarithmic curve. Puzzle, on the other hand, can consider the pruning ratio for each layer in a custom manner, which could also consider assigning higher pruning ratios to later layers like SlimGPT does.
>
> Finally, thank you for mentioning LayerNAS, we will also include it as a reference to related literature in the revision.

---

> > ### Comment · Reviewer_Fna8 · 2025-04-01
> >
> > > "Do you think it is beneficial to include the full list of batch sizes as well?"
> >
> > Yes, it will be good to have this included in the result section.
> >
> > > We also included the speed measurements on TP=1 to also present a "fair" comparison to Puzzle-51B for a single GPU setting.
> >
> > I do agree that while it is fair for the comparison, it is unfair that this setting where 70B is hitting other issues you are reporting ~5x improvement in throughput. Just wanted to note this, do not expect new results here.
> >
> > ----------
> > Based on the rebuttal provided, my questions and associated concerns have been addressed. I will retain my current score based on this.

---

> > > ### Author Response · Authors · 2025-04-02
> > >
> > > We thank reviewer Fna8 for their response,
> > >
> > > *"I do agree that while it is fair for the comparison, it is unfair that this setting where 70B is hitting other issues you are reporting ~5x improvement in throughput."*
> > >
> > > We included the single-GPU comparison to provide a helpful comparison for practitioners who may encounter such constraints. However, we agree that models should primarily be compared under their optimal settings. This is why, throughout the paper—including in the Abstract and in the "Throughput comparison" paragraph in Section 5—we consistently report a 2.17× speedup, rather than emphasizing the ~5× improvement seen in the TP=1 setting. We also agree with you that the issues with this comparison should have been made more explicit and will ensure they are clearly stated in the revision.
> > >
> > > Additionally, we will include the batch sizes in the revised version, as discussed.

---

### Official Review · Reviewer_PN1H · 2025-03-09

**Overall Recommendation:** 2

**Summary:**

The paper introduces Puzzle, a hardware-aware framework that optimizes LLM inference efficiency using neural architecture search (NAS), blockwise local knowledge distillation (BLD), and mixed-integer programming. The authors demonstrate its effectiveness with Puzzle-51B, a 51B-parameter model derived from Llama-3.1-70B, achieving 2.17× inference speedup on a single H100 GPU while retaining 98.4% accuracy despite training on only 45B tokens.

**Claims And Evidence:**

Yes

**Essential References Not Discussed:**

[1]Frantar, Elias, and Dan Alistarh. "Sparsegpt: Massive language models can be accurately pruned in one-shot." International Conference on Machine Learning. PMLR, 2023.

[2]Xia, Mengzhou, et al. "Sheared LLaMA: Accelerating Language Model Pre-training via Structured Pruning." The Twelfth International Conference on Learning Representations.

**Experimental Designs Or Analyses:**

Although comparisons were conducted on some common benchmarks, the advantages of the proposed method are not clearly demonstrated. The paper lacks comparisons with widely-used compression techniques, such as SparseGPT[1] and Sheared LLaMA[2], which limits the ability to assess its relative effectiveness and innovation.

[1]Frantar, Elias, and Dan Alistarh. "Sparsegpt: Massive language models can be accurately pruned in one-shot." International Conference on Machine Learning. PMLR, 2023.
[2]Xia, Mengzhou, et al. "Sheared LLaMA: Accelerating Language Model Pre-training via Structured Pruning." The Twelfth International Conference on Learning Representations.

**Methods And Evaluation Criteria:**

This paper compresses a 70B model to 51B, achieving a very limited compression rate. Although it attains a 2x inference speedup, the performance still falls short of the original level even after additional training. Furthermore, there are no compression results for a 7B model, significantly limiting its practical value.

**Other Comments Or Suggestions:**

None

**Other Strengths And Weaknesses:**

**Weaknesses**:
1. The distillation-based approach for model compression lacks innovation and new insights, as it relies on well-established techniques without introducing novel methodologies or deeper understanding.

2. The experiments only focus on compressing a 70B model to 51B, which has limited practical value. The additional training cost further diminishes the trade-off, making it less appealing for real-world applications.

3. The paper lacks discussion and comparisons with other compression methods, such as **low-rank approximation** and **unstructured/structured sparsity**, which are critical for evaluating the proposed method's competitiveness and effectiveness in the broader context of model compression.

**Questions For Authors:**

See Weaknesses

**Relation To Broader Scientific Literature:**

None

**Theoretical Claims:**

None

---

> ### Author Rebuttal · Authors · 2025-03-30
>
> Thank you for your constructive feedback,
>
> *"no compression results for a 7B model"*
> *"experiments only focus on compressing a 70B model to 51B"*
>
> We demonstrate Puzzle's robustness by applying it 11 times with varied constraints, datasets, and budgets:
>
> (1) 4 derivatives of Llama-3.1-70B (including Puzzle-51B).
>
> (2) 6 derivatives of Llama-3.3-8B.
>
> (3) A derivative of Llama-3.3-70B: "Puzzle-49B" (Sec. 5).
>
> Moreover, post-submission, we applied Puzzle in two additional scenarios:
>
> (4) A 253B derivative of Llama-3.1-405B, constrained for a single H100 node at 1.5X latency, retaining 99.5\% of parent performance (benchmarks: MMLU, MT-Bench, MMLU-Pro, HumanEval, Arena Hard).
>
> (5) A novel 50B+ Mamba-hybrid derivative, constrained for RTX 5090 with 1M context length, retaining 99.94\% parent performance (benchmarks: MMLU, MMLU-Pro, GSM8K, HellaSwag).
>
> We'll include these in the revision to highlight Puzzle's practical value.
>
> *"Although it attains a 2x inference speedup, the performance still falls short of the original level even after additional training."*
>
> We believe a 98\% accuracy retention is impressive. Moreover, while Puzzle-49B already retains high accuracy, we show that a lightweight alignment phase further boosts its performance, leading it to outperform its parent model, Llama-3.3-70B, at 105.5\% relative accuracy.
>
> *"The additional training cost further diminishes the trade-off, making it less appealing for real-world applications."*
>
> Indeed 45B tokens might be expensive for some users, even if it is much lower than the trillions of tokens necessary to train LLMs from scratch. However:
>
> (1) GKD is meant to squeeze extra performance when budget allows. Puzzle derivatives remain competitive even without GKD (Table 14, Appendix F.3), retaining 96.5\% or 90\% parent performance without GKD.
>
> (2) While we didn't mention it in the paper, even a *significantly* shorter GKD training may suffice for a substantial increase in performance. After 3.7B tokens, Puzzle-51B reached 98.8\% parent performance on MMLU and MT-Bench (MMLU 79.38, MT-Bench 8.96). Puzzle-49B, after 8.68B tokens GKD (pre long-context KD in Sec. 5), reached 99.63\% parent performance (80.73 MMLU, MT-Bench 8.87). Even just 2.9B tokens GKD recovered 98.47\% for Puzzle-49B (MMLU 80.72, MT-Bench 8.675).
>
> We agree that it is important to include these results in the revision to clarify how GKD length can be adjusted based on the available budget.
>
> *"compresses a 70B model to 51B, achieving a very limited compression rate."*
>
> As noted in the paper, we argue that categorizing models by parameter count alone (50B vs. 70B) is less meaningful. Real-world choices should depend on hardware, budget constraints, and usage profiles (sequence length, batch size). Hypothetically, a good and resource-efficient 80B model that is faster than an 8B model is preferable to it. We note that even for parameter reduction alone, our accuracy retention remains significant.
>
> *"comparisons with other compression methods, such as low-rank approximation"*
>
> The Puzzle framework is complementary to techniques such as structured sparsity and low-rank approximation, and can be incorporated within its search space. Thus, these techniques enhance rather than compete with Puzzle.
>
> Still, we agree comparisons with methods like [1] and [2] would indeed benefit the paper. We've initiated such evaluations for the revision. Instead of [1], we've chosen Wanda [3], a newer structured sparsity method with good results, which we hope you find acceptable.
>
> Below are preliminary results comparing Puzzle, Wanda, and low-rank approximation. Wanda pruned Llama-3.1-70B (2:4 structured sparsity) targeting similar speedups as Puzzle-51B. The low-rank approximation resembles [4], with subsequent distillation.
>
> | Model             | MMLU  | MT-Bench | Average Accuracy | Accuracy Preserved |
> |-------------------|-------|----------|------------------|--------------------:|
> | Puzzle-51B       | 80.20 | 8.99     | 85.05            | 99.49              |
> | Wanda            | 72.99 | 8.39     | 78.44            | 92.23              |
> | Low-rank         | 72.87 | 8.01     | 76.05            | 88.96              |
> | Llama-3.1-70B    | 81.66 | 8.93     | 85.48            | 100                |
>
> For Wanda [3], which doesn't include additional training, distillation post-pruning yielded marginal gains (MMLU 73.69; MT-Bench unchanged).
> We are also working to evaluate Sheared Llama [2] as you suggested.
>
> Finally, we're exploring integrating these methods within Puzzle, demonstrating their complementary strength, which we'll aim to present clearly in the revision.
>
> [3] Sun et al. A Simple and Effective Pruning Approach for Large Language Models, ICLR
>
> [4] Khodak et al. Initialization and Regularization of Factorized Neural Layers, ICLR
>
> *"The distillation-based approach...lacks innovation and new insights..."*
>
> We respectfully disagree; limited space prevents elaboration but we are happy to clarify if needed.

---

### Official Review · Reviewer_bqJm · 2025-03-12

**Overall Recommendation:** 3

**Summary:**

The paper proposes a NAS pipeline for pruning a pre-trained large language model. The search space includes pruning the attention heads for the attention module and pruning FFN columns (intermediate size) for the FFN module. The pipeline includes three pieces: 1) blockwise local distillation: by training each pruned module with distillation loss to recover the original module output, this part produces a library of pruned module. 2) block scoring: the pruned modules are evaluated by their quality relative to the original modules 3) searching the best combination of pruning strategies: by formulating the pruning problem as a mixted-integer programming problem (constraint is the desired efficiency after pruning), we can search for the best set of pruned modules based on their scores.

Afterwards, the pruned model is trained with distillation loss end-to-end again to additional heal the gap. The entire procedure utilizes 45B training tokens and is able to get a 51B model out of llama-3.1-70B-Instruct with almost no performance lost.

**Claims And Evidence:**

Yes, both the speedup and the accuracy of the model support the efficacy of the method.

**Essential References Not Discussed:**

N/A

**Experimental Designs Or Analyses:**

The main expeirmental design makes sense. There are not much ablations/analysis in the main paper though, which can be an area for further improvement.

**Methods And Evaluation Criteria:**

The accuracy evaluation setup makes sense, but the author only evaluates the method on one model (llama-3.1-70B), which raises concern on its generalizability.

**Other Comments Or Suggestions:**

TLDR: put more ablation results in the main text instead of leaving them in the appendix

With a lot of design choices and experiments covered in the paper, I think the paper lacks a clear outline of the core research question/methodology it is investigating and the takeaways from the experiments. As the paper proposes a new pruning methodology for a fairly well-known pipeline, the focus should be on the ablation of various method design choices. I saw the author had a good amount of ablation in the appendix and the author should select some of them to be put in the main text along with relevant discussion to provide the reader more insights on what is important for pruning the model.

**Other Strengths And Weaknesses:**

Strengths
- Comprehensive evaluation and strong empirical results.

Weakness
- There are several metrics considered for the replace-1-block score, but I don't find where/if the author conducts ablation studies on them and which one is the best (there is only one analysis in the appendix which doesn't fully address this question).
- The global distillation phase requires a large amount of tokens. 45B tokens in total is not a small number and hinder the applicability of the method.
- The method is only tested on llama-3.1-70B, it is unclear if the methodology can transfer to other model sizes/families.

**Questions For Authors:**

N/A

**Relation To Broader Scientific Literature:**

There are two branches of LLM pruning papers: structural pruning and unstructural pruning. This paper falls under the category of structural pruning and proposes a systematic NAS method to prune the model which is not found in previous literature. The methodology (recovery error, distillation) isn't super new but in combined showed promising performance. The main strength of this paper is its empirical success, where most of the unstructured pruning have either worse performance or an insufficient evaluation setup.

**Theoretical Claims:**

Not applicable for this paper.

---

> ### Author Rebuttal · Authors · 2025-03-30
>
> Thank you for your feedback and appreciation of the ablation studies,
>
> *"There are several metrics considered for the replace-1-block score, but I don't find where/if the author conducts ablation studies on them and which one is the best"*
>
> Appendix F.1.4. examines the impact of different replace-1-block scores. In short, for general use, it is best to use KL divergence as the metric. We'll make sure to state this conclusion more clearly in the revision.
> We also found that if a particular downstream task is prioritized, using data similar to that task for block scoring will produce results that outperform the KL divergence *on this specific task*, but underperform KL Divergence solutions across different tasks. See the Half-MMLU experiment in Appendix F.1.4. for more details. Additionally, LM Loss always underperforms KL divergence as a block score (as shown in Figure 7).
>
> *"the author only evaluates the method on one model (llama-3.1-70B), which raises concern on its generalizability."*
>
> The paper shows the application of the Puzzle method in crafting 11 different models with various constraints, datasets and budgets to show the robustness of the method:
>
> (1) 4 derivatives of Llama-3.1-70B:
> a) Puzzle-51B,
> b) 2 other 51B derivatives with a different BLD token budget (Appendix F.1.3., Table 9),
> c) A Puzzle derivative from a different dataset (Gutenberg dataset, Appendix F.1.2., Table 8),
> d) A limited search space variant (Appendix F.1.5., Table 11)
>
> (2) 6 derivatives of Llama-3.3-8B:
> a) A "coupled BLD" derivative (appearing both in the main paper at Table 5 and in Appendix F.1.4., Table 7),
> b) 2 derivatives with LM loss block scoring (Figure 7 in Appendix F.1.4.),
> c) 2 derivatives with decoupled BLD (Figure 7 in Appendix F.1.4.),
> d) A derivative with a downstream, "Half-MMLU" block score (Table 10 in Appendix F.1.4.)
>
> (3) A derivative of Llama-3.3-70B: "Puzzle-49B" (Section 5).
>
> We agree that applying Puzzle to produce a large variety greatly strengthens the paper. That is why, after submission, we applied Puzzle in two additional scenarios with different parent models:
>
> (4) Using Puzzle, we created a 253B derivative of Llama-3.1-405B with Puzzle constraints to fit a single H100 node at a 1.5X latency. The resulting model retains 99.5% of the parent model performance (averaged on MMLU, MT-Bench, MMLU-Pro, HumanEval and Arena Hard).
>
> (5) Experimenting with a novel Mamba-hybrid model (consisting of more than 50B parameters) as a parent, we crafted a Puzzle derivative while constraining it to fit a single RTX 5090 with a 1M context length. The resulting model retains a 99.94% of the parent's performance (average on MMLU, MMLU-Pro, GSM8K and HellaSwag).
>
> We will make sure the revision includes these examples to emphasize Puzzle's robustness across a variety of constraints and models.
>
>
> *"The global distillation phase requires a large amount of tokens. 45B tokens in total is not a small number and hinder the applicability of the method."*
>
> We agree 45B tokens might not be an applicable budget for every user, even if it is much lower than the trillions of tokens necessary to train LLMs from scratch. However:
>
> (1) GKD is meant to squeeze extra performance when budget allows. Puzzle derivatives remain surprisingly competitive even without GKD (see Table 14 in Appendix F.3., where Puzzle derivatives retain 96.5% or 90% of the parent's performance without applying GKD at all).
>
> (2) While we didn't mention it in the paper, even a *significantly* shorter GKD training may suffice for a substantial increase in performance, and we will emphasize this in the revision. After only 3.7B tokens invested in GKD, Puzzle-51B had already recovered 98.8% of its parent's performance on MMLU and MT-Bench (MMLU 79.38, MT-Bench 8.96). Puzzle-49B underwent a GKD of only 8.68 tokens (prior to the long context KD described in Section 5), at which stage it already recovered 99.63% of its parent's performance (80.73 MMLU, MT-Bench 8.87). Even after just 2.9B tokens for GKD, Puzzle-49B had already recovered 98.47% (MMLU 80.72, MT-Bench 8.675). We agree that it is important to include these results in the revision to clarify how GKD length can be adjusted based on the available budget.
>
>
> *"the author had a good amount of ablation in the appendix and the author should select some of them to be put in the main text"*
>
> Thank you for your positive feedback, the ablation studies were of utmost importance for us to objectively conclude the best configurations for using Puzzle in a robust way. We intend to move several ablations into the main body.
>
> In particular, since our core question in the ablation studies was to find the best configuration for applying Puzzle, we believe the ablations presented in F.1.1., F.1.3. and F.1.4. are the most important for practitioners (with F.1.2. also contributing to data preparation, if space constraints in the main body allow us to add it as well). What is your opinion on this selection?

---

> > ### Comment · Reviewer_bqJm · 2025-04-02
> >
> > I think the rebuttal provides further evidence of the generalizability of the method in terms of both the target model and training size. I will thereby increase my score to 3.
> >
> > Regarding the ablation, I would love to have the BLD section (F.1.3 and F.1.4) be brought to the main text since the BLD seems to be one major methodology novelty of the paper and I think a clean presentation on these two sections can allow the reader better understand how it should be applied.

---

> > > ### Author Response · Authors · 2025-04-02
> > >
> > > We thank reviewer bqJm for their response and score increase,
> > >
> > > *"Regarding the ablation, I would love to have the BLD section (F.1.3 and F.1.4) be brought to the main text since the BLD seems to be one major methodology novelty of the paper and I think a clean presentation on these two sections can allow the reader better understand how it should be applied.*"
> > >
> > > We agree with your suggestion and will move these sections into the main text in the revision.

---

### Official Review · Reviewer_GBJ9 · 2025-03-13

**Overall Recommendation:** 3

**Summary:**

This paper is concerned with model compression, which aims to compress the scales of LLMs. This paper proposes a NAS framework named Puzzle to conduct easy-to-achieve NAS. The Puzzle framework firstly trains decoupled blocks for each layer via block-wise local distillation, then searches best-fit plan for architecture, and finally uptrains the searched architecture for preserved performance. The experimental results show that the NAS-based architecture is more efficient than original architecture and is competitive with the original model in a wide range of tasks.

**Claims And Evidence:**

The claims are supported by clear and convincing evidence. However, I still have several concerns:
1) The comparison is sub-optimal and lacks critical baselines concerning about distillation, making the claims seem to be overclaimed.

**Essential References Not Discussed:**

N/A

**Experimental Designs Or Analyses:**

The experimental designs and analyses are valid. However,I still have several concerns:
1) It would be much better to integrate the above-mentioned baselines as performance references.

**Methods And Evaluation Criteria:**

The proposed methods are clear and the evaluation criteria is mostly adequate. However, I still have several concerns:
1) Key baselines are missing, which 1) uses distillation on a fully random architecture 2) uses distillation on a random-from-block-library architecture.

**Other Comments Or Suggestions:**

N/A

**Other Strengths And Weaknesses:**

N/A

**Questions For Authors:**

N/A

**Relation To Broader Scientific Literature:**

N/A

**Theoretical Claims:**

The proofs for theoretical claims are correctly justified.

---

> ### Author Rebuttal · Authors · 2025-03-31
>
> Thank you for your feedback and suggestion,
>
> *"Key baselines are missing, which 1) uses distillation on a fully random architecture 2) uses distillation on a random-from-block-library architecture."*
>
> We conducted evaluations on the baselines you suggested, namely (1) a fully random architecture and (2) a random-from-block-library architecture. Both were constructed to adhere to the same speed constraints as Puzzle-51B.
> Additionally, we extended (1) with an extra baseline: (3) using Llama-3.1-70B itself with randomized weights (an experiment we dub "Parent-Randomized"). This baseline explores whether increased capacity might have contributed to better performance.
> To ensure fairness, we allocated the same 10B token budget for training each of these models:
>
> | Model                       | MMLU  | MT-Bench | Average Accuracy | Relative to Llama-70B |
> |----------------------------|:-----:|:-------:|:---------------:|:--------------------:|
> | Puzzle-51B (10B tokens)    | 79.7  | 8.89    | 84.3           | 98.61%              |
> | Random-from-block-library  | 66.02 | 8.2     | 74.01          | 86.58%              |
> | Fully Random               | 23.13 | 0.89    | 16.015         | 18.73%              |
> | Parent-Randomized          | 23.42 | 0.95    | 16.46          | 19.25%              |
> | Llama-3.1-70B              | 81.66 | 8.93    | 85.48          | 100%                |
>
> We will include these baselines in the revision.
>
> We believe the "Random-from-block-library" experiment is particularly informative, as it highlights the value of the MIP algorithm in selecting high-quality blocks from the block library. Additionally, our paper examines another baseline to the MIP algorithm in Appendix F.2.2., where we use a greedy algorithm to select blocks from the block library.

---

### Decision · Program_Chairs · 2025-05-01

**Decision:**

Accept (poster)

**Comment:**

This paper presents Puzzle, a distillation-based NAS framework for optimizing LLMs for efficient inference. It achieves strong empirical results, including a 2.17× throughput speedup on an H100 GPU with minimal accuracy loss. The method is well-engineered, robust across multiple model sizes, and grounded in practical deployment constraints.

While some reviewers noted limited theoretical novelty and missing comparisons, the authors provided thorough rebuttals, added key baselines, and committed to improving the revision. Given the overall strength of the work and its practical relevance, it is a solid contribution to the field.